# Meta-Sparsity: Learning Optimal Sparse Structures in Multi-task Networks through Meta-learning

## Abstract

This paper presents meta-sparsity, a framework for learning model sparsity, basically learning the parameter that controls the degree of sparsity, that allows deep neural networks (DNNs) to inherently generate optimal sparse shared structures in multi-task learning (MTL) setting. This proposed approach enables the dynamic learning of sparsity patterns across a variety of tasks, unlike traditional sparsity methods that rely heavily on manual hyperparameter tuning. Inspired by Model Agnostic Meta-Learning (MAML), the emphasis is on learning shared and optimally sparse parameters in multi-task scenarios by implementing a penalty-based, channel-wise structured sparsity during the meta-training phase. This method improves the model's efficacy by removing unnecessary parameters and enhances its ability to handle both seen and previously unseen tasks. The effectiveness of meta-sparsity is rigorously evaluated by extensive experiments on two datasets, NYU-v2 and CelebAMask-HQ, covering a broad spectrum of tasks ranging from pixel-level to image-level predictions. The results show that the proposed approach performs well across many tasks, indicating its potential as a versatile tool for creating efficient and adaptable sparse neural networks. This work, therefore, presents an approach towards learning sparsity, contributing to the efforts in the field of sparse neural networks and suggesting new directions for research towards parsimonious models.

## 1 Introduction

Model compression in Deep Learning (DL) is a process that aims at reducing the size and complexity of neural network models while maintaining their performance. In the current technological trend, model compression has become essential in DL due to its role in enabling complex neural networks to operate efficiently on devices with constrained resources by substantially improving memory usage and computation requirements. It facilitates the practical application of DL models in real-world scenarios, fostering sustainability and enhancing accessibility on a broader scale (Deng et al. (2020)). It is possible to reduce the size and complexity of a model by leveraging techniques such as Neural Architecture Search (NAS)(Ren et al. (2021)), model value quantization, model distillation(Hinton et al. (2015)), low-rank factorization (Sainath et al. (2013)), parameter sharing (Desai & Shrivastava (2023)), sparsification (Hoefler et al. (2021)), and many more. This work focuses on two of these techniques, i.e., parameter sharing in the form of hard parameter sharing in multi-task learning and sparsification. However, our emphasis extends beyond simply compressing models. We combine the concept of model sparsification with Multi-Task Learning (MTL), viewing sparsification as a technique for selecting optimally shared features in a multi-task setting. This methodology enables the strategic distribution of these features across diverse tasks during MTL, enhancing the joint learning process.

Sparsification[1] in DL pertains to transforming a densely connected neural network or a dense model to a sparse one, where a considerable proportion of the parameters (typically model weights) are set to zero. This reduction of active weights leads to less computational load and memory usage during both the training and

---

[1] *Note:* In the context of this work, *sparsification* should not be confused with *pruning*, which is also a widely used term in model compression. This is because both are different concepts; pruning is a technique that can help to achieve sparsification by systematically removing certain elements; sparsification is a much broader concept that can be reached through various methods, including but not limited to pruning. So the words sparsification and pruning are **not** used interchangeably.

inference phases. Additionally, sparsification not only highlights the importance of key features (sparsity-driven feature selection) but also significantly enhances the model's ability to generalize to new data (Hoefler et al. (2021)). When implementing sparsity in Deep Neural Network (DNN), three key aspects must be considered: (i) identifying what elements should be made sparse, (ii) determining when to induce sparsity, and (iii) deciding on the method i.e., how to achieve sparsity. The article by Hoefler et al. (2021) offers a comprehensive survey that thoroughly explores these factors related to sparsity, among various other elements. Several elements can be sparsified in a DNN, like the parameters (or weights), neurons, filters in convolution layers, and heads in attention layers. Sparsification can be scheduled post-training, during training, or as sparse training, as illustrated in Figure 1. Further details on these methods are provided in the subsequent text. Various techniques can be employed to attain sparsity, including methods like magnitude pruning based on thresholds, sparsification based on input-output sensitivity, penalty-driven approaches, variational methods, and many more. In this work, we emphasize sparsifying the model parameters during the training by applying penalty-based (regularization-based) sparsification methods.

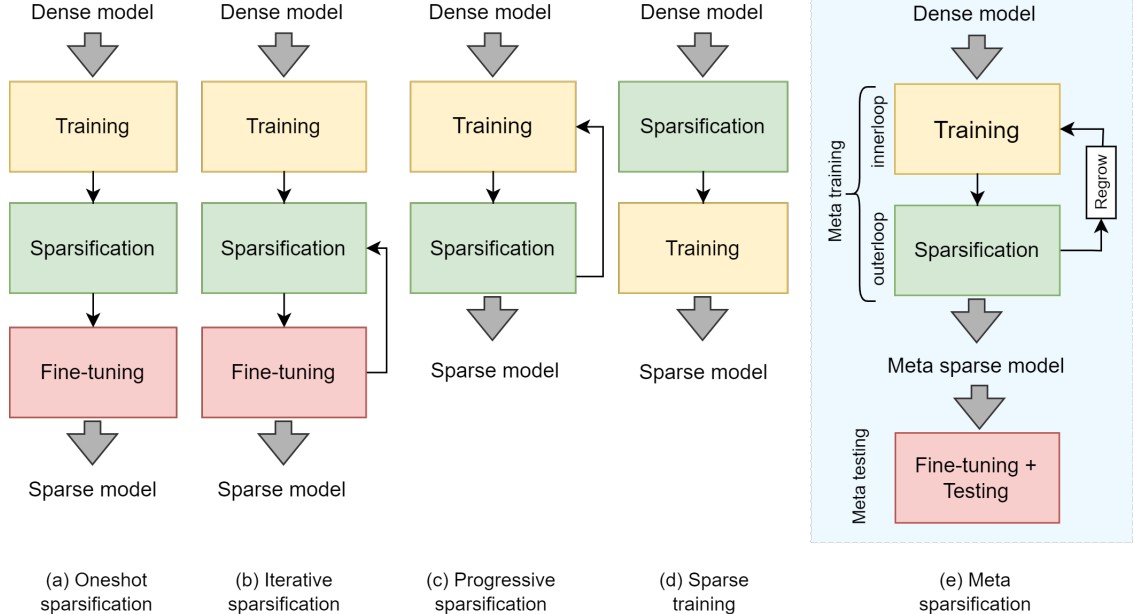

Figure 1: This figure illustrates (a-d) a few common approaches to achieving sparse models and (e) the proposed approach of meta-sparsification.

Various sparsity-inducing strategies have been developed to facilitate the transformation of a dense model into a sparse model (Hoefler et al. (2021)). However, a vital question that arises before implementing these strategies is determining when and how to apply sparsification. Figure 1 broadly illustrates some of these sparsity scheduling techniques as discussed by Hubens (2020). The one-shot sparsification (Figure 1(a)) is a fundamental technique that sparsifies an already trained dense model until a desired sparsity is achieved and then fine-tunes the sparse model. Fine-tuning allows the model to relearn. It helps to re-distribute the weights across the remaining parts of the network, ensuring that performance does not suffer significantly. Figure 1(b) shows iterative sparsification where sparsity is induced gradually over many steps (Han et al. (2015); Liu et al. (2017)). Only a handful of weights are eliminated at each iteration, and the network is fine-tuned to restore performance. This process is repeated multiple times. Zhu & Gupta (2017) presented the concept of automated progressive (or gradual) sparsification (Figure 1(c)) in their work, which is very similar to iterative sparsification. While both methods incrementally introduce sparsity into a dense network, automated gradual sparsification is different because it sparsifies the weights during the training of the model, allegedly claiming that it eliminates the necessity for subsequent fine-tuning. Figure 1(d) illustrates sparse training, where a sparse model is trained, and while training, it tries to retain the sparsity. This is often referred to as sparse-to-sparse training (Dettmers & Zettlemoyer (2019); Evci et al. (2020)), while the others Figure 1(a-c) are referred to as dense to sparse training.

In general, any sparsification involves the use of a few hyper-parameters that control the degree of sparsity, like thresholds in magnitude pruning, sparsification budget (or target), the regularization parameter in penalty-based sparsity approach, the dropout rate (probability), learning rate in gradual pruning, and several others. Often, approaches like grid search, random search, heuristic methods, etc, are employed to determine the optimal values of these crucial hyper-parameters. We introduce a method to learn these hyper-parameters that induce sparsity through meta-learning. This work, therefore, proposes meta-sparsification, broadly depicted in Figure 1(e). In this work, we refer to it as *learning to learn sparsity*, or *learned sparsity*, or *meta-sparsity*. To be more precise, it is learning to learn the sparsity controlling hyper-parameter, thereby learning the sparsity pattern.

We begin with a dense multi-task model, which shares some layers i.e., a shared encoder or backbone between many tasks. To achieve a clear understanding, we begin by answering the three essential questions related to sparsification. (i) *What to sparsify?* - we aim to sparsify the parameters or weights of the model, primarily the parameters of the shared layers of a multi-task network. We apply channel-wise structured sparsity to the shared layers to achieve optimal shared representations across all the tasks. Structured sparsity is a coarse-grained approach that considers the architecture of the dense network while sparsifying i.e., for a Convolutional Neural Network (CNN); it involves zeroing out entire filters, neurons, or channels, basically a structured block of parameters. (ii)*how to achieve sparsity?* (i.e., the method) - we apply penalty-based (regularization-based) structured sparsity to the backbone of a multi-task network. The sparsification is applied in a setting inspired by *gradient-based meta-learning* e.g., Model Agnostic Meta Learning (MAML) (Finn et al. (2017)) to learn the meta-sparsity patterns across all the tasks. Meta-learning (Thrun & Pratt (2012); Huisman et al. (2021)), commonly referred to as 'learning to learn,' involves developing algorithms that learn from a variety of tasks and then apply the accumulated knowledge to enhance future learning of new or similar tasks. (iii) *when to sparsify?* - sparsity is applied while training the network; this is also known as *dynamic sparsity*. It refers to an approach where the sparsity pattern[2] changes over time during training, adapting to the evolving data and training requirements, while static sparsity refers to fixed sparsity patterns throughout training. The broader objective of this work is to explore learning the sparsity patterns across multiple learning episodes or tasks and to develop a meta-sparse model during the meta-training stage that can potentially be fine-tuned for the same or unseen tasks in the meta-testing stage (as shown in Figure 1(e)).

Now, the question is why this approach is applied in a multi-task context. Traditional MAML often involves tasks or learning episodes that are very similar or homogeneous; for example, all are image classification tasks (Hospedales et al. (2022); Upadhyay et al. (2023b)). Because of this similarity, tasks may exhibit consistent sparsity patterns, hindering the meta-model's ability to accommodate a wider variety of tasks with different sparsity requirements. MTL, in contrast, enables the simultaneous learning of multiple, diverse tasks. Therefore, we aim to identify robust sparsity patterns that are suitable for various diverse tasks by using sparsity in multi-task scenarios and learning sparse behavior across tasks through meta-learning. The main contributions of this work are as follows -

- *Learned sparsity framework called Meta-sparsity*: We introduce a novel approach that allows for the dynamic learning of sparsity in DNN. This is a significant shift from heuristic-based sparsity control to a more dynamic, task-aware method. Our method is versatile, being agnostic to model type, task, and sparsity type, allowing for wide applicability.

- *Comprehensive evaluation:* We rigorously assess our approach across a variety of tasks using two publicly available datasets, showcasing its effectiveness. To provide comparative insights, we compare these results to those obtained from models using no sparsity and fixed-sparsity (i.e., the fixed value of sparsity hyper-parameter) in single-task and multi-task settings.

- *Robustness validation:* The robustness of our meta-sparse models is demonstrated by their capacity to perform satisfactorily on previously unseen tasks during meta-testing, showcasing their potential adaptability.

---

[2]*sparsity pattern* refers to the specific arrangement or layout of non-zero parameters within a model architecture, for example, a CNN when structured sparsity is applied. In the context of channel-wise structured sparsity, this pattern identifies the active (non-zero) channels within the network layers from the zeroed-out (inactive) ones. These patterns highlight the selective engagement of certain parts of the network while others remain dormant

- *Versatility in sparsity types:* Although this work focuses on channel-wise structured sparsity, we also validate the efficacy of our approach on unstructured sparsity, demonstrating its broad utility.

- *Direction for future research:* We conclude with a discussion on potential future directions and open questions to encourage more investigation in the areas of model sparsity.

This paper is structured as follows: Section 2 delves into the related work, setting the context and background for this study. Section 3 describes the methodology of the proposed approach. Section 4 details the experimental setup, while Section 5 presents the results, performance analysis, and a thorough discussion of the findings. Finally, Section 6 concludes the work and suggests directions for future research.

## 2 Related work

This section focuses on research works that have employed the notion of learning diverse entities (i.e., hyper-parameters, optimizers, loss functions, and many more) through the utilization of black-box models or alternative methodologies. We additionally delve into the utilization of sparsity in deep learning and subsequently focus our attention on the application of sparsity in MTL networks. Finally, we position our work within the existing literature.

**What can be learned?:** The concept of learning to learn, commonly known as meta-learning, has been a learning paradigm of great interest in research for many years. This field has significantly advanced since the seminal works of Schmidhuber (1987); Bengio et al. (1991); Thrun & Pratt (2012), leading to a wide range of applications. These algorithms aim to achieve generalization by learning from experiences, with the extent of generalization depending on the accumulation of meta-knowledge. In simple words, this generalization can be achieved by learning to learn parameters, hyper-parameters, loss functions, architectures, optimizers, and many more, which constitutes meta-knowledge. Many excellent studies have been conducted in this field of study; in order to set a context for our work, we will attempt to highlight and briefly discuss some of them.

We will first begin with *learning to learn parameter initialization.* Transfer learning centers on the concept of providing generalized initial parameters for the downstream task, enabling the fine-tuning of the new task with relatively fewer data samples. A possible approach to find the best initial parameters is to utilize meta-learning algorithms to train the parameters across different tasks, acquiring meta-knowledge that can aid in rapid adaption to new tasks. The article by Finn et al. (2017) presented MAML an algorithm that optimizes its parameters to facilitate adaptability and quick learning across a diverse range of tasks by providing a set of initial parameters. Reptile (Nichol & Schulman (2018)) is also a similar algorithm that is mathematically similar to first-order MAML, that learns the initialization of a network. First-order MAML (FOMAML) (Nichol et al. (2018)) and Almost No Inner Loop (ANIL) (Raghu et al. (2019)) are simplifications of MAML that provide computational advantages compared to MAML. The comprehensive survey by Huisman et al. (2021) provides detailed insights into other related work within the field of meta-learning.

While training neural networks, one of the most tedious tasks is hyperparameter tuning. This is because it involves a trial-and-error process over a vast, complex search space, and often, the hyperparameters are very use-case specific; there is no guarantee of finding the best solution, and each trial can be computationally expensive and time-consuming. That is why *learning to learn hyperparameters* is critical. Some works by Li et al. (2017); Xiong et al. (2022); Chen et al. (2023); Subramanian et al. (2023) focused on learning the learning rate or learning the learning rate schedules to train a deep learning model. Many articles learn to adapt all the hyperparameters, including the learning rate. For example, the article by Baik et al. (2020) adaptively generated per-step hyperparameters to improve the performance of the model. Bohdal et al. (2021) presented an approach for hyperparameter optimization by leveraging evolutionary strategies to estimate meta-gradients. Another similar work by Franceschi et al. (2018) offered a structured bi-level programming approach where the outer level updated hyper-parameters while the inner loop focused on task-specific learning or loss minimization. The approach presented by Franceschi et al. (2018) closely aligns with our work, with the primary distinction being our specific emphasis on learning structured sparsity within the framework of Multi-Task Learning (MTL) by employing MAML.

The articles by Wortsman et al. (2019); Gao et al. (2021); Bechtle et al. (2021); Gao et al. (2022b); Raymond et al. (2023a;b) aim to learn parametric loss functions, thereby *learning to learn a loss function*. On similar grounds, several works focus on *learned optimizers* or black-box or parametric optimizers, like the research by Bengio et al. (1991); Andrychowicz et al. (2016); Wichrowska et al. (2017); Lv et al. (2017); Li & Malik (2017); Shen et al. (2020); Harrison et al. (2022); Metz et al. (2022; 2020); Gao et al. (2022a). The learning-to-learn concept is also used in the field of neural architecture search (Elsken et al. (2019)) to achieve optimal architectures. This idea of *learned architectures* is discussed by Lian et al. (2019); Shaw et al. (2019); Elsken et al. (2020); Ding et al. (2022); Rong et al. (2023); Schwarz & Teh (2022). Numerous studies in existing literature apply learning to learn concepts across various domains, such as reinforcement learning, attention learning, and neural memory learning, among others. However, this paper will not delve into each of these applications, as it is not intended to be a comprehensive review of the field. Instead, our primary objective is to highlight distinct insights and developments within a more limited domain of learning sparsity. Overall, it can be summarized that the black-box or parameterized models are increasingly replacing traditional white-box aspects of deep learning, such as loss functions, optimization algorithms, automated architecture search, and others. Building on this concept, this paper presents an approach to *learning to learn sparsity* using meta-learning.

Sparsification is one of the approaches for model compression in Deep Neural Networks (DNNs). It does not just reduce the complexity of the model but also leads to significant gains in performance, computation, and energy efficiency, all while strategically selecting key features that contribute to these gains. Hoefler et al. (2021) presented an elaborate study of sparsity in deep learning. They give an extended survey of works in this domain based on the types of sparsity, what can be sparsified, when to sparsify, and how to introduce sparsity. Some other works that discuss model sparsity or pruning are Zhu & Gupta (2017); Gale et al. (2019). Our work is in line with the area of *learned sparsity*, in contrast to works achieving sparsity using fixed hyper-parameters. Most of the works in the domain focus on adaptive weight or parameter pruning (Han et al. (2015)), a technique to induce sparsity in a network. Kusupati et al. (2020) proposed a dynamic sparsity parameterization (STR) technique that used the sparse threshold operator to achieve sparsity in DNN weights by learning layer-specific pruning thresholds. Unlike the static (fixed) pruning methods, where the sparsity hyperparameter is fixed once pruning is performed, this method dynamically adjusts the sparsity hyperparameter during training, thereby determining the optimal sparsity pattern.

Another similar work by Carreira-Perpinan & Idelbayev (2018) introduced the learning-compression method, a two-stage process used by the algorithm to prune the network. In the learning phase, the weights are optimized e.g., reducing loss, while in the compression phase, network pruning is done using $l_0$ or $l_1$ constrain along with fixed pruning hyperparameters. Instead of manually adjusting the pruning rates of each layer, Zhou et al. (2021b) introduced ProbMask, a technique that leverages probability to determine the importance of weights across all layers and allows automatic learning of weight redundancy levels based on a global sparsity constraint. A combination of structured and unstructured sparsity is employed by Zhou et al. (2021a), resulting in N:M fine-grained sparsity wherein each group of M consecutive weights of the network, there are at most N weights that have non-zero values. So, this method does weight pruning based on the N:M budget. Many other works focus on adaptive pruning of networks or learning structured sparsity (Wen et al. (2016); Meng et al. (2020); Liu et al. (2018); Lee et al. (2021); Wang et al. (2021); Upadhyay et al. (2023c)); however, none have applied meta-learning for learning to learn parameter sparsity. The articles contributed by Wen et al. (2016); Meng et al. (2020); Deleu & Bengio (2021) provide comprehensive insights into the concept of structured sparsity in deep neural networks. The idea of structured sparsity constitutes an essential aspect of our paper, and we draw significant inspiration from these works.

**Sparsity in multi-task learning:** The domain of Multi-Task Learning (MTL) confronts two significant challenges. Firstly, as the number of tasks increases, the total number of trainable parameters also increases. Secondly, it is crucial to efficiently group similar tasks or effectively share features across tasks during training to prevent adverse information transfer among tasks, which could potentially impact the performance of certain tasks. Sparsity is used in the literature as one of the solutions to overcome these problems in multi-task settings. For example, Kshirsagar et al. (2017) constructed a regularization term to jointly optimize the task-specific parameters while also learning shared group structures (or parameters). Similarly, Gonçalves et al. (2016) in their work introduce Multi-task Sparse Structure Learning (MSSL) for joint estimation

of per-task parameters and their relationship structure parameters using Alternating Direction Method of Multipliers (ADMM) algorithm (Boyd et al. (2011)). Argyriou et al. (2006), also assumed that tasks are related and used $l_1 - l_2$ regularization with the combined loss for efficient multi-task feature learning. Obozinski et al. (2010); Chen et al. (2009) used different sparse and non-sparse regularizations to learn the low-dimensional subspace, which is shared by all the tasks. Sun et al. (2020a) proposed a method for iterative magnitude pruning of multi-task model parameters, which was inspired by the lottery ticket hypothesis (Frankle & Carbin (2019)). Their method induces unstructured $l_1$ type sparsity (sparse masks for weight matrices) by training sub-nets for multiple tasks until convergence.

**Positioning our work:** In the literature stated earlier, we looked into numerous ideas, such as learning to learn an optimizer, loss function, initial parameters, hyper-parameters, and many more. Building upon these discussions, we introduce an approach that utilizes meta-learning to facilitate the process of *learning to learn sparsity*. From the view of MTL, the previously mentioned studies induce sparsity in multi-task models by utilizing fixed values for regularization parameters, pruning thresholds, or sparsification budgets. In contrast to employing a trial-and-search methodology for the fixed hyper-parameter, the present study emphasizes acquiring the ability to learn a generalized sparsity parameter across many tasks. This approach can also be viewed as one of the strategies for optimal feature sharing among tasks within a multi-task framework facilitated by learned structured sparsity. To the best of our knowledge, this research direction has not been previously explored, offering a unique contribution to the fields of sparsity and MTL.

## 3 Methodology

In this section, we outline our proposed methodology. First, we will look into the conventional MTL, meta-learning with a focus on MAML, and group (structured) sparsity in order to establish a strong foundation for *meta sparsity*. This foundational discussion sets the stage for understanding the nuances of our approach.

### 3.1 Multi-task Learning

Multi-task learning (Caruana (1997)) is a very well-established learning paradigm wherein the aim is to jointly learn or train multiple related tasks. The underlying theory here is that the tasks help each other to learn better due to inductive transfer between the tasks, leading to improved performance and better generalization. A successful MTL can be achieved by establishing a balanced sharing between tasks, such that there is a positive transfer of information. This primarily depends on the parameter sharing approach used for MTL, which, according to Crawshaw (2020), can be either hard parameter sharing or soft parameter sharing. Hard parameter sharing is a result of the shared architecture, while soft parameter sharing can be achieved by applying constraints on the model parameters. These concepts, along with various research efforts in this area, are explored in a comprehensive survey on MTL by Crawshaw (2020). This work employs a very simple and conventional multi-task architecture, demonstrating hard-parameter sharing see Figure 2.

Consider a task distribution p(T), from which, say, N non-identical yet related tasks are sampled that can be trained simultaneously in a multi-task setting, say, $\mathcal{T} = \{T_1, T_2, ..., T_N\}$. The objective of MTL is to train a multi-task model, say $f(\Theta)$ so that a combined loss ($\mathcal{L}$) of the task-specific losses $\{l_{T_1}, l_{T_2}, \ldots, l_{T_N}\}$ is minimized, such that the optimal parameters are given by the Equation 1,

$$\Theta^* = \arg\min_{\Theta} \mathcal{L}(f(\Theta)) = \arg\min_{\theta_b, \theta_i \in \Theta} \mathcal{F}_c(l_{T_i}(f_i(\theta_b, \theta_i))), \qquad i = 1, 2, \ldots N \tag{1}$$

where $\Theta = \{\theta_b\} \cup \bigcup_{i=1}^{N}\{\theta_i\}$, indicating that the set of multi-task parameters $\Theta$ comprises $\theta_b$, the parameters of the shared architecture or backbone, and $\theta_i$ the parameters specific to each of the N tasks. $f_i$ denotes the model designated for a particular task, consisting of a shared backbone ($\theta_b$) along with layers specific to the task $\theta_i$. $\mathcal{F}_c$ represents the function that combines or, more precisely, balances the N task-specific losses. This function may be a straightforward sum of losses, a weighted average, or any other tailored function designed to suit the specific requirements of the use case. To maintain fair learning across all tasks and to avoid any one task from dominating the learning process, loss balancing is essential in MTL. Several approaches to combine single-task losses to ease the multi-task optimization are discussed in Crawshaw (2020).

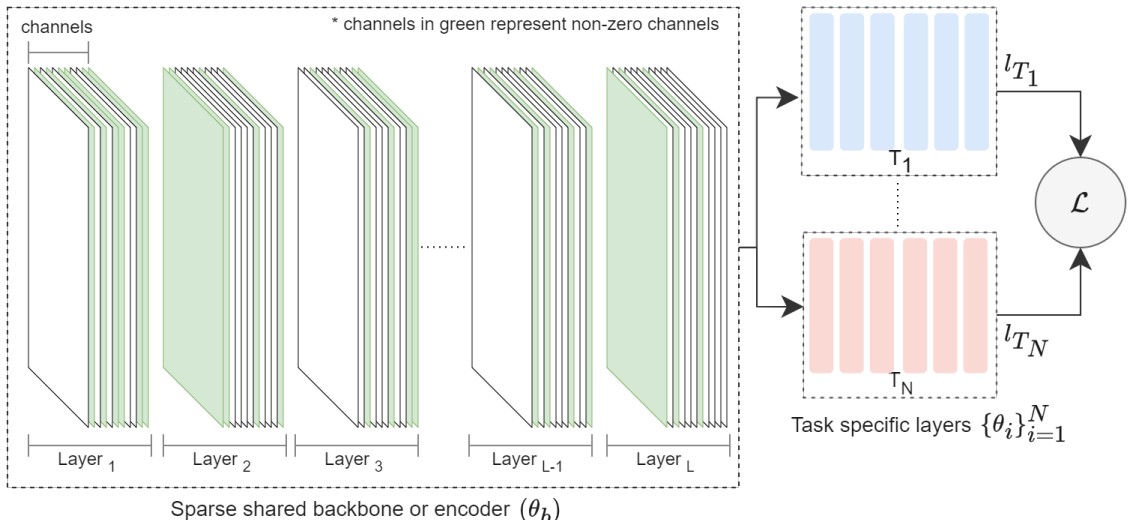

Figure 2: A schematic of the multi-task architecture used in this work (inspired by Upadhyay et al. (2023c)).

In this work, we employ the uncertainty weighting approach given by Kendall et al. (2018). The core idea of this approach is to optimize learning outcomes in multi-task contexts by prioritizing or weighing the tasks based on their uncertainty. It can mathematically be represented as,

$$\mathcal{L} = \mathcal{F}_c(l_{T_1}, l_{T_2}, \ldots, l_{T_N}) = \sum_{i=1}^{N} \left( \frac{1}{2\sigma_i^2} \cdot l_{T_i} + \log \sigma_i \right) \tag{2}$$

where $\sigma_i$ is a learnable noise parameter for $i^{th}$ task. This combined loss $\mathcal{L}$ undergoes backpropagation, where gradients of the loss are computed with respect to all model parameters $\Theta$, encompassing both shared and task-specific parameters. These gradients are then used to update $\Theta$ to minimize $\mathcal{L}$ by finding the optimized parameters, thereby enhancing the performance of all tasks.

## 3.2 Meta-learning

Meta-learning, often known as 'learning to learn' (Thrun & Pratt (1998); Baxter (1998)), is a learning algorithm that leverages past learning experiences to enhance the performance of a new task. In conventional Machine Learning (ML), an algorithm is designed to acquire knowledge from a given dataset to perform a specific task. However, meta-learning goes a step further; it involves algorithms that can evaluate their learning process, analyze their performance across various tasks, and leverage this insight to learn new tasks more efficiently. Meta-learning algorithms can be classified into three categories, i.e., (i)model-based, (ii)metric-based, and (iii)optimization-based or gradient-based meta-learning. These classifications are thoroughly discussed in several survey papers, including works by Huisman et al. (2021); Tian et al. (2022); Hospedales et al. (2022), and many more, which provide in-depth analyses and comparisons of methodologies within each category. Out of these, in this work, we focus on optimization or gradient-based meta-learning algorithms, particularly MAML (Finn et al. (2017)). MAML facilitates fast and efficient adaptation of a DNN to new and distinct tasks, using only a limited amount of training data. This is accomplished by optimizing a set of initial parameters that are highly adaptable and enable quick fine-tuning via minimal gradient updates. This approach improves model generalizability and performance and has been thoroughly tested in a variety of settings (tasks), such as classification, regression, and reinforcement learning. In this work, we extend its application beyond similar or homogeneous tasks to dense prediction tasks like segmentation, depth

estimation, and others by incorporating MTL in the form of multi-task learning episodes (Upadhyay et al. (2023a) of heterogeneous tasks [3].

The MAML framework works with many tasks, also defined as learning episodes, and progresses through two hierarchical levels (or loops): the inner loop, focusing on learning from individual tasks through rapid adaptation to training data, and the outer loop, which aggregates knowledge from numerous tasks to enhance and accelerate the adaptation process. These levels are formalized in the form of a bilevel optimization problem in MAML in the meta-training stage, which is addressed hereafter. After meta-training, the 'learning to learn' stage, new tasks can be introduced in the meta-testing stage, also known as the 'adaptation stage.' Consider, for N tasks or learning episodes sampled from task distribution p(T), $\mathcal{L}_{meta}$ and $l_{T_i}$ be the meta (i.e., outer) and $i^{th}$ task-specific (i.e., inner) loss functions respectively. Along the same lines, as formulated in Section 3.1, let $\Theta_{meta}$ be the meta parameters, while $\theta_i$ be the parameters for $i^{th}$ task. For training, the dataset for N tasks can be expressed as $\mathcal{D} = \{(D_{sup_1}, D_{query_1}), \ldots, (D_{sup_N}, D_{query_N})\}$, where $D_{sup}$ stands for support set and $D_{query}$ stands for the query set such that $D_{sup_i} \cap D_{query_i} = \emptyset$. The bi-level optimization in MAML, where one optimization problem involves another as a constraint (Hospedales et al. (2022); Huisman et al. (2021); Finn et al. (2017)), can be written as follows :

$$\Theta^*_{meta} = \underset{\Theta_{meta}}{\arg\min} \sum_{i=1}^{N} \mathcal{L}_{meta}(\theta^*_i, \Theta_{meta}, D_{query_i}) \qquad (outer\ loop) \qquad (3)$$

$$where, \quad \theta^*_i \equiv \theta^*_i(\Theta_{meta}) = \underset{\theta}{\arg\min}\ l_{T_i}(\theta, \Theta_{meta}, D_{sup_i}) \qquad (inner\ loop) \qquad (4)$$

So, $\Theta_{meta}$ can be generalized initialization of model parameters; it can also be hyperparameters such as learning rate, regularization parameter, or parameterized loss functions. This work aims to concurrently 'learning to learn' the structured sparsity regularization parameter along with the multi-task model parameters that offer a suitable initial point for model training in the meta-testing stage. Further details are provided in Section 3.4.

### 3.3 Group sparsity

Sparsity in DL can be introduced in two broad ways: as model sparsity, affecting the model structure, or as ephemeral sparsity, impacting individual data examples, as discussed by Hoefler et al. (2021). As already discussed in Section 1, this work explores model sparsity, especially regularization(penalty)-based model weight sparsification. In the context of how sparsity is introduced to model parameters, two distinct modes emerge - structured and unstructured sparsity. Unstructured sparsity (also known as fine-grained sparsity), the more straightforward method, eliminates the least significant weights based on chosen criteria, leading to irregular sparse patterns. In contrast, structured sparsity (also known as group sparsity) adopts a more holistic strategy by targeting the elimination of whole groups of parameters. In this context, a *group* refers to various structural elements within the model, such as channels, filters, neurons, tailored blocks, or attention heads, each representing a distinct set of parameters (Hoefler et al. (2021); Liu & Wang (2023)). Unstructured sparsity can significantly reduce the model size but typically achieves only modest speedups on hardware for dense computations, such as GPUs. Meanwhile, structured sparsity is designed to complement optimized hardware architectures, leading to improvements in computational efficiency. This work primarily concentrates on structured sparsity, utilizing penalty-based techniques for model weight sparsification. Additionally, while our approach centers on structured sparsity, we have also implemented it for unstructured sparsity. The findings and insights are discussed in Section 5.

Suppose the weights of a model are divided into G non-overlapping groups of varying sizes, denoted as $\Theta = \{\theta^1, \theta^2 \ldots \theta^G\}$. $\mathcal{L}(f(\Theta))$ represents the loss function, where $f(\Theta)$ represents the model with parameters $\Theta$. The objective function for the regularization-based group sparsity (group lasso) given by Yuan & Lin

---

[3]As defined by Upadhyay et al. (2023b), tasks are considered homogeneous when they share similar objective, and if the objectives differ, they are classified as heterogeneous. For example, if the objective of all the tasks is image classification, they can be referred to as homogeneous tasks.

(2005) can be expressed as -

$$\Theta^* = \arg\min_{\Theta} \quad \mathcal{L}(f(\Theta)) + \underbrace{\lambda \sum_{g=1}^{G} \sqrt{n^g} \, ||\theta^g||_2}_{l_1 - l_2 \text{ norm regularization term:} \mathcal{R}} \qquad where \quad \theta^g \in \Theta \tag{5}$$

where $\mathcal{R}$ is the regularization term that zeros out entire groups of parameters denoted by $\theta^g$. Here, $n^g$ is the number of elements in the $g^{th}$ group, such that $n^g > 1 \, \forall g$, it is multiplied to achieve normalization of the regularization term across groups of varying sizes. The $l_2$ norm of the parameter group is represented by the term $||\theta^g||_2 = \sqrt{\sum_j (\theta_j^g)^2}$, here $\theta_j^g$ is the parameter at index $j$ for group $g$. In Equation 5, the regularization parameter is represented by $\lambda$, which controls the degree of sparsity by balancing the trade-off between model complexity and data conformity. We aim to learn this hyperparameter using MAML in meta-sparsity over multiple tasks. Given the regularization term $\mathcal{R}$ combines the $l_1$ norm of $l_2$ norms across groups, it is often referred to as $l_1 - l_2$ norm in this work.

To optimize the composite loss function in Equation 5, proximal gradient methods can be applied since the regularization term $\mathcal{R}$ is non-differentiable (Parikh et al. (2014); Bach et al. (2012)). The proximal gradient descent updates the parameters based on the gradient of the differential part of the composite loss function given by

$$\Theta_{t+1} \leftarrow prox_{\alpha\mathcal{R}}(\Theta_t) = prox_{\alpha\mathcal{R}}(\Theta_t - \alpha\nabla_\Theta \mathcal{L}(f(\Theta_t))). \tag{6}$$

Here, $prox_{\alpha\mathcal{R}}$ is the proximal operator, and $\alpha$ is the learning rate. Since $\Theta$ is divided into $G$ disjoint groups, the proximal operator can be written as,

$$prox_{\alpha\mathcal{R}}(\Theta) = (prox_{\alpha\mathcal{R}}(\theta^1), \ldots, prox_{\alpha\mathcal{R}}(\theta^g), \ldots, prox_{\alpha\mathcal{R}}(\theta^G)) \tag{7}$$

In the case of group sparsity-inducing regularization, this proximal operator can be computed and has a closed form as discussed by Combettes & Wajs (2005); Hastie et al. (2015); Scardapane et al. (2017); Deleu & Bengio (2021), given by the equation,

$$prox_{\alpha\mathcal{R}}(\theta^g) = \begin{cases} \left[1 - \dfrac{\alpha\lambda\sqrt{n^g}}{||\theta^g||_2}\right]\theta^g & ; ||\theta^g||_2 > \alpha\lambda\sqrt{n^g} \\ 0 & ; ||\theta^g||_2 \leq \alpha\lambda\sqrt{n^g} \end{cases} \tag{8}$$

So, if the $l_2$ norm of a group of parameters is less than (or equal to) a threshold depending on the size of the group and regularization parameter, the entire group of parameters is zeroed out, thereby enforcing group sparsity.

### 3.4 Proposed approach - Meta sparsity

Now that we have established the concepts of MTL, meta-learning, and group sparsity, we will discuss the proposed approach of meta-sparsity or learned sparsity in this section. Now, in the context of this work, to precisely answer the three questions related to sparsity- (i) *what to sparsify?*: the objective is to sparsify the backbone parameters i.e., $\theta_b$, also known as the shared layers of the multi-task network. (ii) *how to sparsify?*: apply group sparsity (i.e., Equation 5) to the backbone parameters $\theta_b$ only. The channels of the convolution layer weight matrix are treated as different groups, i.e., $\theta_b^l(:, c, :, :)$ which represents one group consisting of the weights of $c^{th}$ channel of the $l^{th}$ layer. This is symbolically illustrated in Figure 2. (iii) *when to sparsify?*: this proposed approach is inspired by meta-learning i.e., learning to learn sparsity; therefore, the sparsification is done during the meta-training phase in the outer loop. This approach, in general, can be viewed as a *learning-sparsification* iterative methodology. It involves episode-specific learning at the inner loop (Equation 4) and parameter sparsification at the outer loop (Equation 3) of MAML in an iterative manner. Therefore, similar to meta-learning, meta-sparsity also has two phases: the meta-learning phase and the meta-testing (adaptation) phase. An overview of the meta-training and testing stages is given in the table below. Adding one more question to the ones above, (iv) *why sparsification?*: besides model compression, the main intention behind applying structured sparsity is efficient feature selection across multiple tasks.

Furthermore, sparsification promotes generalization, an important aspect when incorporating new yet related tasks into the multi-task framework.

Since MAML is an episodic learning approach (Hospedales et al. (2022); Finn et al. (2017)), it necessitates several learning episodes, which essentially involve single-task learning. In this study, we extend the concept of learning episodes to include multi-task learning episodes (Upadhyay et al. (2023a)), encompassing all possible combinations of the N tasks. For N tasks i.e., $\mathcal{T} = \{T_1, T_2, \ldots, T_N\}$, the learning episodes $\mathcal{E}$ used for meta-training are an ensemble of single-task and multi-task learning episodes given by the power set of $\mathcal{T}$, such that,

$$\mathcal{E} = 2^{\mathcal{T}} \setminus \{\emptyset\} = \{E_1, E_2, ..., E_N\} = \{T_1, T_2, .., T_1T_N, .., T_1T_2T_N, ...\} \tag{9}$$

The training procedure is inspired by MAML(Finn et al. (2017)) but with two notable distinctions. First, the training process involves a multi-task model being trained on single-task and multi-task learning episodes to utilize sparsity patterns specific to each context; Second, in addition to the model parameters, the hyperparameter that induces sparsity, denoted as $\lambda$, is also meta-learned, which results in a meta-sparse multi-task model.

|  | META-TRAINING |  | META-TESTING |
|---|---|---|---|
|  | **Inner loop** *(learning)* |  | **Fine-tuning** *(similar to MTL)* |
| Objective: | reduce the episode-specific loss |  | *Begin with - Meta sparse multi-task model* |
|  | i.e. $\mathcal{L}_{E_i}$ |  | Three possibilities: |
| Train: | episode-wise training of multi- |  | (i) Fine-tuning on all the same N tasks |
|  | task model with sparse backbone |  | as meta-training. |
| Optimize: | multi-task model parameters |  | (ii) Add new task and fine-tune only |
|  | for every learning episode. |  | on the new $(N+1)^{th}$ task |
|  | **Outer loop** *(sparsification)* |  | (iii) Add new task and fine-tune on all |
| Objective: | reduce the meta-loss |  | the (N+1) tasks. |
|  | i.e. $\mathcal{L}_{meta}$ |  | *(maintain sparsity while fine-tuning)* |
| Train: | meta-train the multi-task model |  |  |
|  | on all learning episodes. |  | **Testing** |
| Optimize: | multi-task model parameters, |  | Use an independent test set to evaluate |
|  | regularization parameter |  | the performance of the meta-sparse |
|  | (inducing channel-wise sparsity) |  | multi-task model |
| Outcome: | Meta sparse multi-task model |  |  |

In CNN, the weight tensor is organized as a 4D array, with dimensions representing the number of filters, channels(or depth), height, and width. When considering structured sparsity, the specific structures that can be regularized include filters, channels, filter shapes, or a customized block of parameters. In this work, we employ channel-wise structured sparsity to the shared backbone of the multi-task architecture. Several previous studies, including those by Wen et al. (2016); Deleu & Bengio (2021); Upadhyay et al. (2023c), among others, have focused on channel-wise sparsity across a range of applications, mainly because of two reasons. First, this approach generates smaller dense networks by eliminating redundant channels, effectively compressing the network. Furthermore, these optimized networks offer greater hardware benefits, especially for GPUs, as they can be more easily accelerated. The emphasis on channel-wise sparsity improves both computing efficiency and the suitability of models for deployment on devices with limited resources.

Consider a simple and conventional multi-task architecture with a shared encoder or backbone: a CNN and N task-specific networks (a.k.a heads) connected to the backbone. In this case the multi-task model parameters $\Theta = \{\theta_b\} \cup \bigcup_{i=1}^{N} \{\theta_i\}$ where $\theta_b$ are the shared parameters and $\theta_i$ are the task-specific parameters for $i^{th}$ task. Since the group sparsity is applied to the channels of convolution layers of the backbone network, the meta-optimization objective as per Equation 3 and 4 can be expressed along with group sparsity (Equation 5) as,

$$\Theta^*_{meta} = \underset{\Theta_{meta}}{\arg\min} \sum^{N_b} \mathcal{L}_{meta}(\Theta^*_{E_i}, \Theta_{meta}, D_{query_{E_i}}) + \lambda \sum_{g=1}^{G} \sqrt{n^g} ||\theta^g_{b_{E_i}}||_2, \quad s.t. \ E_i \in \mathcal{E} \quad (outer\ loop) \tag{10}$$

$$where, \quad \Theta^*_{E_i} \equiv \Theta^*_{E_i}(\Theta_{meta}) = \underset{\Theta_{E_i}}{\arg\min} \mathcal{L}_{E_i}(\Theta_{E_i}, \Theta_{meta}, D_{sup_{E_i}}). \qquad (inner\ loop) \tag{11}$$

$$Also, \quad ||\theta^g_{b_{E_i}}||_2 = \sum_{l=1}^{L} \sum_{c=1}^{C_l} ||\theta^l_{b_{E_i}}(:,c,:,:)||_2. \qquad (l_2 \ norm \ of \ a \ group \ i.e. \ a \ channel) \qquad (12)$$

Here, $L$ is the number of convolution layers, and $C_l$ is the number of channels in the $l^{th}$ layer. $N_b$ represents the total number of batches of data in batch-wise training of the model. $\lambda$ is the learnable regularization parameter. $E_i$ stands for the learning episode see Equation 9 Here, $\mathcal{L}_{E_i}$ is the episode-specific loss if the learning episode has a single task, then $\mathcal{L}_{E_i} = l_{T_j}$ such that, $T_j \in \mathcal{T}$. But if it is a multi-task episode, say the $\mathcal{L}_{E_i} = \mathcal{F}_c(l_{T_a}, l_{T_b}, l_{T_c})$, where $T_a, T_b, T_c \in \mathcal{T}$ (see Equation 1). To avoid the potential divergence of $\lambda$ towards negative infinity, a Softplus function is applied to $\lambda$, which ensures that it remains positive throughout the optimization process. The Softplus function, defined as $\text{Softplus}(x) = \frac{1}{\beta} \log(1 + \exp(\beta \cdot x))$, acts as a smooth approximation to the ReLU function and effectively constrains $\lambda$ to positive values.

The Equation 10 and 11 illustrate the process of bi-level optimization. The inner loop trains the multi-task model for various task combinations, while the outer loop finds the meta-optimized multi-task network parameters and generates a channel-sparse backbone network. This process involves meta-learning the channel sparsity and model parameters. The full algorithm of meta-training is outlined in Algorithm 1.

---

**Algorithm 1** Meta-sparsity (training)

---

**Require:** Sample N tasks from a task distribution p(T), say $\mathcal{T} = \{T_1, T_2, ... T_N\}$
**Require:** MTL model with parameters, $\Theta = \{\theta_b\} \cup \bigcup_{i=1}^{N}\{\theta_i\}$ ▷ where $\theta_b$ = shared parameters and $\theta_i$ = task-specific parameters
**Require:** Initialize meta parameters $\Theta_{meta}$ ▷ dense model
**Require:** Single & multi-task episodes: $\mathcal{E} = \{E_1, E_2, .., E_{2^N-1}\}$
   **while** not converged **do**
      **for** $b_s = 1$ **to** no. of batches of $D_{sup}$ and $D_{query}$ **do** ▷ outer loop
         Randomly sample an episode $E_i$
         $\mathcal{G} \leftarrow [\ ]$ ▷ Using [ ] to denote an empty list
         Initialize $\Theta_{E_i} \leftarrow \Theta_{meta}$
         **if** $\Theta_{E_i}$ has sparse groups and regrow probability $r_p > 0$ **then**
            - Select groups for regrowth (randomly or some criteria))
            - Initialize the selected groups
         **end if**
         **for** $\kappa$ inner updates **do** ▷ inner loop
            **if** $E_i$ contains $n$ tasks such that, $n > 1$ **then**
               $\mathcal{L}_{E_i} = \mathcal{F}_c(l_{T_1}, l_{T_2}, \ldots, l_{T_n})$ ▷ multi-task loss
            **else**
               $\mathcal{L}_{E_i} = l_{T_j}$ ▷ single task loss
            **end if**
            Calculate gradients, $g = \nabla_{\Theta_{E_i}} \mathcal{L}_{E_i}(\Theta_{E_i}, \Theta_{meta}, D_{sup_{E_i}})$
            Update $\Theta^*_{E_i} \leftarrow \Theta_{E_i} - \alpha_{in}\, g$ ▷ $\alpha_{in}$ is the learning rate for adaptation stage
         **end for**
         For the query set, calculate $G_i = \nabla_{\Theta^*_{E_i}} \mathcal{L}_{meta}(\Theta^*_{E_i}, \Theta_{meta}, D_{query_{E_i}})$
         Accumulate gradients, $\mathcal{G} \leftarrow \mathcal{G} \cup \{G_i\}$
      **end for**
      Calculate meta gradients, $\mathcal{G}_{meta}$, as the average of all accumulated gradient
      Update $\Theta^*_{meta} \leftarrow prox_{\alpha_{out}}(\Theta_{meta} - \alpha\, \mathcal{G}_{meta})$ ▷ $\alpha_{out}$ is the learning rate for meta stage
   **end while**

---

Within the meta-training inner loop, the concept of regrowth can also be introduced, which involves reinstating previously pruned (or sparsified) parameters, as highlighted by (Hoefler et al. (2021)). This strategy is employed because regrowth aids in overcoming excessive sparsification enables the model to adjust to new data patterns, and ensures a balance between achieving sparsity and maintaining performance (Sun et al. (2023); Zhang et al. (2024)). Moreover, the literature suggests that when regrowth is iteratively used along with sparsity (or pruning), it contributes to discovering better parsimonious network architectures. The extent of parameter regrowth is modulated by a hyper-parameter $r_p$ that governs the probability of regrowth.

Regrowth can be implemented in the inner loop during meta-training in order to enhance episode-specific learning in this work. We did not apply regrowth to all trials to avoid influencing the outcome of meta-sparsity. However, we included some results for comparison in Section 5 (under the subheading - Regrowing the parameters).

In meta-testing, we have the meta sparse multi-task model, which is further fine-tuned (as discussed in Figure 1(e)). A new task may be introduced during this stage to evaluate the generalizability of the meta-model. During fine-tuning, the sparsity pattern of the backbone parameters is maintained, and only the task-specific heads of the new tasks are trained. This is similar to standard multi-task training (Equation 1). Once we have the fine-tuned multi-task model, its performance is evaluated on the test set.

In this study, we primarily emphasize meta-learning structured $(l_1 - l_2)$ sparsity for the shared layers in a multi-task setting. However, through extensive experiments, we demonstrate that our approach is not limited to structured sparsity alone; it is equally adept at meta-learning other forms of penalty-based sparsity, such as $l_1$ (unstructured) sparsity. For this work, we limit the focus of our work only to regularization (or penalty) based sparsity-inducing approaches. However, we are certain that the proposed approach has broader applications and can facilitate learning sparsity in various elements, including weights, activations, or any aspect that can be adjusted via a hyperparameter. Furthermore, the learning process involves various hyperparameters, such as learning rate, momentum, activation functions, and dropout probability, which can also be optimized through meta-learning. However, this paper focuses on meta-learning the model parameters and the regularization hyperparameter. This focused approach will allow us to thoroughly examine the sole influence of learning sparsity on model performance.

## 4    Experimental setup

In this section, we elaborate on the experimental framework designed to investigate the effectiveness of our proposed meta-sparse multi-task models. Here, we provide an overview of the datasets chosen for multi-task learning, describe the structure of the multi-task network used, and outline the specific evaluation metrics employed for each task. Additionally, we outline the types of experiments designed to assess the effectiveness of our proposed methodology.

Table 1: A table containing the loss functions and evaluation metrics for the various tasks for both the datasets: NYU-v2 and CelebAMask-HQ. In the table a downward arrow ($\downarrow$) represents that a lower value is better while an upward arrow ($\uparrow$) represents a higher value is better. Also, IoU stands for intersection over union and MAE stands for mean absolute error.

| Dataset | Tasks | $(T)$ | Loss Functions | Evaluation Metric |
|---|---|---|---|---|
| NYU-v2 | Semantic segmentation | $T_1$ | cross-entropy loss | IoU ($\uparrow$) |
| | Depth estimation | $T_2$ | Combination of errors in depth gradient and surface normal(Hu et al. (2019)) | MAE ($\downarrow$) |
| | Surface Normal estimation | $T_3$ | Inverse cosine similarity | cosine similarity ($\uparrow$) |
| | Edge detection | $T_4$ | huber loss(Paul et al. (2022)) | MAE ($\downarrow$) |
| CelebA Mask-HO | Semantic segmentation | $T_1$ | cross-entropy loss | IoU ($\uparrow$) |
| | Binary classification (attributes- male, smile, big lips, high cheekbones, wearing lipstick, bushy eyebrows) | $T_2 - T_7$ | binary cross-entropy loss | Accuracy ($\uparrow$) |

In this work, two widely recognized and publicly accessible datasets, the NYU-v2 dataset (Nathan Silberman & Fergus (2012)) and CelebAMask-HQ dataset (Lee et al. (2020)), are used, hereafter referred to as NYU and CelebA, respectively in this paper. Table 1 details the various tasks considered for each dataset, the respective loss functions, and evaluation metrics (see Appendix or supplementary material for details). We have adopted the standard loss functions (and metrics) as utilized in previous works such as Sun et al. (2020b); Upadhyay et al. (2023a) and many others, to primarily focus on assessing the impact of sparsity on task performance and avoiding performance enhancements that could arise from the use of complex, custom-designed loss functions. Overall, in the NYU dataset, all the tasks are pixel-level tasks, segmentation is pixel-level classification, and the rest are pixel-level regression. While in the celebA dataset, there is only one pixel-level (classification)

task, and the rest are image-level (classification) tasks. This set of tasks was chosen explicitly to extensively evaluate the performance of the proposed approach across a wide range of task combinations, proving their effectiveness and adaptability in many contexts.

As already discussed in Section 3, a very standard multi-task network architecture is chosen, which has a backbone network i.e., the layers shared by all the tasks, and task-specific heads or layers are connected to the output of the backbone, see Figure 2. We selected the dilated ResNet-50 (Yu et al. (2017)) architecture as our backbone network due to its resilience to sparsity; even when sparsity leads to entire layers having zero parameters, the residual connections within the network ensure continuity by effectively propagating values forward. This characteristic maintains the network's structural integrity and facilitates uninterrupted information flow, making it an ideal choice for our experiments. For the dense prediction tasks, a deeplab-v3 network (Chen et al. (2017)) is employed as a task-specific network, and for the binary classification task, a two-layer fully connected network is used. In the past, similar network designs and architecture were also used by Kendall et al. (2018); Liebel & Körner (2018); Upadhyay et al. (2023a), and many more. For a fair comparison of results and to maintain consistency across evaluations, all the experiments in this work use the same architecture, loss functions, metrics, train-validation-test split, and hyperparameters. Models are trained using NVIDIA A100 Tensor Core GPUs, which have 40 GB of onboard HBM2 VRAM. To assess the reliability of the model, we conducted five replications of each experiment using distinct random seeds. The findings are presented in terms of the mean and the standard deviation. To ensure reproducibility, the source code can be accessed at: https://github.com/PLACEHOLDER TO THE GIT REPOSITORY

The following are the types of experiments designed and analyzed in this work-

- Experiments *without* group sparsity (using the dense model, $\lambda = 0$)

    1. Single task learning - one model for each task
    2. Multi-task learning - employ a multi-task model for different task combinations

- Experiments *with* group sparsity (applying channel-wise structured sparsity on the backbone layers)

    1. Single task learning + sparse backbone; Fixed sparsity (i.e., $\lambda$ is fixed)
    2. Multi-task learning + sparse backbone; Fixed sparsity (i.e., $\lambda$ is fixed)
    3. Multi-task learning + meta-sparse backbone; Learnable sparsity (i.e., $\lambda$ is meta-learned)

During meta-testing, keeping the sparse backbone from meta-training, the performance of the tasks under the three settings is compared:

    1. Meta-testing or fine-tuning on the same tasks as meta-training.
    2. Add a new task, and fine-tune only the new task.
    3. Add a new task to the meta-training task and fine-tune all the tasks.

Note that the single and multi-task experiments, both without and with fixed sparsity do not use meta-learning for parameter initialization. Meta-learning is only used in meta-sparsity experiments. Additionally, all the multi-task experiments are performed for 3-4 task combinations. Given the computation requirement, not all the $2^N - 1$ combinations for N tasks are evaluated. For the NYU dataset, we examine three multi-task scenarios: (i) the complete set $(T_1, T_2, T_3, T_4)$ encompassing all tasks, (ii) a mixed set of classification and regression tasks $(T_1, T_2, T_3)$ with one task reserved for meta-testing, and (iii) a subset of solely regression tasks $(T_2, T_3, T_4)$, omitting the classification task for meta-testing purposes. Similarly, for the celebA dataset, the following combinations are considered : (i) all the seven tasks i.e., $(T_1 - T_7)$, (ii) all the binary classification tasks i.e., $(T_2 - T_7)$, and (iii) $(T_1, T_2, T_3, T_7)$ other very similar attributes primarily related to the mouth region of the image were added during meta-testing to study the performance.

## 5   Results and Discussion

This section primarily answers the following questions: why multi-task over single-task learning? Is the meta-sparsity approach viable? How do the meta-sparse multi-task models perform compared to the fixed

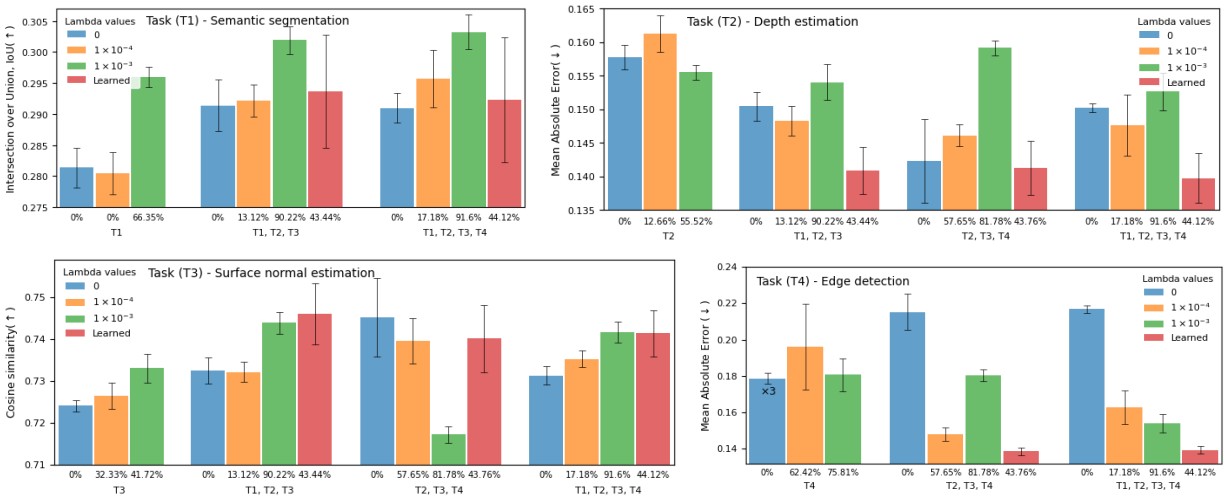

Figure 3: For NYU dataset, task-wise performance comparison of single-task and multi-task no sparse ($\lambda = 0$, in blue), fixed sparsity single and multi-task (for $\lambda = 1 \times 10^{-4}$ in yellow and $\lambda = 1 \times 10^{-3}$ in green) and meta sparsity ($\lambda =$ learned in red i.e., meta-sparsity) experiments. The vertical axis label is annotated with an upward or downward arrow to indicate whether a higher or lower metric value is preferable. The values below each bar represent the percentage of parameter sparsity in the backbone network. The 'x3' on the initial bar for task T4 indicates that the depicted performance metric (MAE) is triple the current value represented by the bar. This notation is employed to simplify the depiction of values that exhibit significant disparities.

sparsity multi-task models? How effectively is the sparse backbone model processing a novel, unseen task? Therefore, this section aims to provide comprehensive insight into the applicability and advantages of the proposed meta-sparsity approach within MTL learning, supported by empirical evidence. Detailed results are provided in the Appendix for further reference.

*Note:* For the fixed lambda experiments, we looked at how different tasks and combinations of tasks respond to the same setting of a sparsity parameter, $\lambda$. This parameter controls the level or amount of sparsity, which is measured in terms of (%) parameter sparsity[4]. We focused on three specific settings of this parameter: $1 \times 10^{-3}$, $1 \times 10^{-4}$, and $1 \times 10^{-5}$ because these values induced sparsity in most of the single-task and multi-task settings.

**Assessing the viability of meta-sparsity:** Figures 3 and 4 illustrate the task-wise performance for the NYU and celebA datasets, respectively. For ease of comparison, the bar charts display the results for single-task learning (both with and without fixed sparsity), multi-task learning (again, with and without fixed sparsity), and meta-sparsity multi-task learning (featuring learned sparsity) across all tasks. For the NYU dataset (Figure 3), for tasks $T_2$ and $T_4$, the meta-sparse multi-task models significantly outperform compared to other model configurations. For task $T_3$, these models achieve performance on par with their counterparts, whereas for task $T_1$, they slightly under-perform relative to the single or multi-task fixed sparsity counterparts, while are equivalent or better than not sparse models. Meta-sparse models surpass their fixed-sparsity counterparts in terms of performance despite having a lower level of parameter sparsity, highlighting a trade-off between performance and sparsity.

For the celebA dataset, meta-sparsity excels for some task combinations. However, specifically for the task combination $T1, T_2, T_3, T_7$, the performance across all the tasks, except for $T_1$ (i.e., segmentation), is undesirable. A probable reason behind this can be that the task ensemble involves classification tasks aimed at significantly distinct attributes. This under-performance can primarily be attributed to task interference, where the meta-sparsity approach faces challenges in identifying optimal features for sharing between tasks. However, the expanded combination of $T_1 - T_7$ leads to improved performance for many of these tasks. This

---

[4]In this work, *parameter sparsity* refers to the proportion of the model's parameters which are zeroed out, represented as a percentage

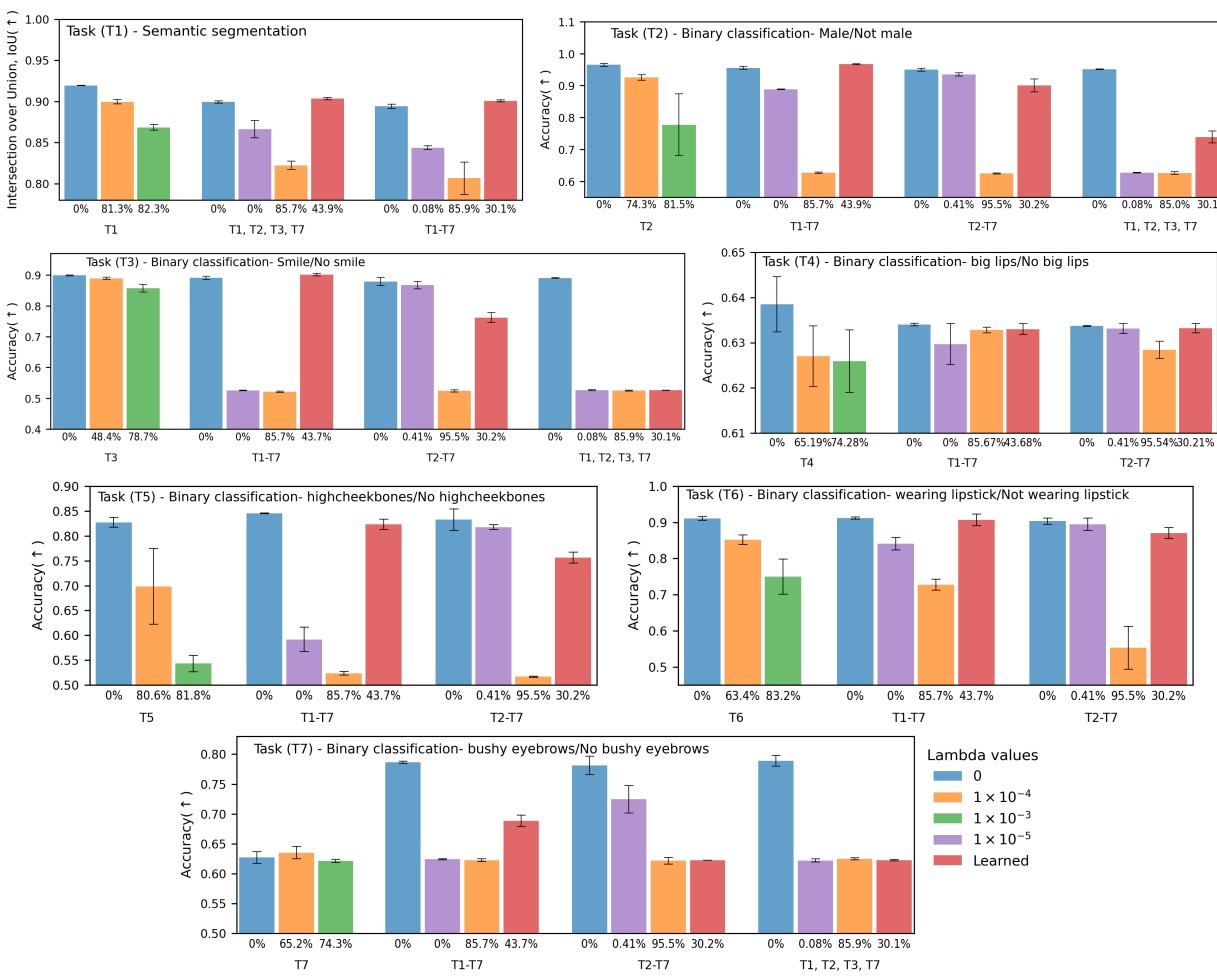

Figure 4: For celebA dataset, task-wise performance comparison of single-task and multi-task no sparse ($\lambda = 0$, in blue), fixed sparsity single and multi-task (for $\lambda = 1 \times 10^{-5}$ in lavender, $\lambda = 1 \times 10^{-4}$ in yellow and $\lambda = 1 \times 10^{-3}$ in green) and meta sparsity (in red) experiments. The vertical axis label is annotated with an upward or downward arrow to indicate whether a higher or lower metric value is preferable.

improvement suggests that more related (similar attributes) tasks improve performance through enhanced feature sharing among tasks facilitated by meta-sparsity. Some exceptions are discussed further in this section.

Note that to enhance the visibility and comparability of the performance differences, we chose a narrow y-axis range for the plots in Figure 3, which may have amplified the visual impact of the variability of the error bars representing the standard deviation. Please refer to Table 1 in the supplementary material (which forms the basis of Figure 3), which quantitatively demonstrates that the standard deviations are consistently low. However, it is evident that for many tasks the variance in the performance of meta-sparsity is slightly greater than the rest (no sparsity and fixed sparsity). One probable reason can be 'Regularization Stochasticity'. It can be defined as the variability in the learned sparsity patterns because of the stochastic nature of (mini-batch) gradient-based optimization and the dynamic penalty-based sparsity applied during training. This stochasticity can result in different paths of convergence and patterns of sparsity when the same experiment is run multiple times; therefore, the variability in the performance. Another factor to consider, that can amplify these stochastic effects, is the random initialization of the strength of the regularization hyperparameter ($\lambda$), which is trainable/learned in meta-sparsity. We sample the initial value of $\lambda$ from a uniform distribution between 0.1 and 1. While this may seem like a small range, it can still lead to significant variations in the convergence paths and the resulting sparsity patterns.

**Comparative performance analysis:** Before evaluating the proposed meta-sparsity approach, we would like to provide an insight into why MTL is preferred over single-task learning. Experiments (Figures 3 and 4) typically show that tasks perform better in a multi-task environment compared to a single-task environment. This observation emphasizes the importance of utilizing multi-task learning episodes. Additionally, the amount of sparsity varies depending on the specific task (in single-task scenarios) and the combination of tasks (in multi-task scenarios). There is no clear trend in sparsity levels that might help to predict how the sparsity of a multi-task model changes in relation to the number of tasks. It can be inferred that sparsity is affected by the nature and interconnection of the tasks involved. Further elaboration on task performance in the celebA dataset is provided in the following subsections.

Interestingly, when analyzing the outcomes for the celebA dataset, we noticed that for the lowest value i.e., $1 \times 10^{-5}$, the experiments, both single and multi-task, show no sparsity at all, and the performance is also similar to the without sparsity setting. While for $1 \times 10^{-3}$ and $1 \times 10^{-4}$, they exhibit very high sparsity (around 85% - 95%) in just one or two epochs, but this comes at the cost of not being trained enough, which negatively affects its performance (not for single tasks). This is the reason behind the poor performance of all the tasks for $\lambda = 1 \times 10^{-3}$ and $1 \times 10^{-4}$, see the Figure 4 (the lavender and yellow bars). In the case of single-task experiments, they perform better since the feature requirement in single-task settings is much simpler (straightforward) than in multi-task settings. From the above analysis, it is logical to infer that other values of $\lambda$, both lower than $1 \times 10^{-5}$ and higher than $1 \times 10^{-3}$, would exhibit similar behaviors. Specifically, values below $1 \times 10^{-5}$ may not cause considerable sparsity, potentially resulting in overfit models without performance advantages, while values above $1 \times 10^{-3}$ may result in sparse models early in training, impeding learning and performance. This conjecture highlights the delicate balance needed to optimize sparsity and model performance when selecting $\lambda$. This fuels the rationale behind the primary aim of this research, which is to learn the optimal sparsity parameter $\lambda$, which in turn is expected to enhance performance.

Also, for the task $T_7$ i.e., bushy eyebrows/ no bushy eyebrows, the performance of meta sparsity is not up to the mark for any of the task combinations. This shortfall could be attributed to the unique nature of the task. Most selected tasks are associated with attributes of the entire image, such as segmentation and gender classification, or specifically relate to the mouth area, including distinctions like smiling/not smiling, having big lips/no big lips, and wearing lipstick/not wearing lipstick; the sparse features dominate towards these tasks. In contrast, task $T_7$ is the only task that concentrates on the eye region, specifically eyebrows, which likely requires features different from those needed for the other tasks, which is why its performance in isolation is much better. Similarly, the semantic segmentation task $T_1$ from the NYU dataset performs satisfactorily in a fixed-sparsity situation but does not reach comparable performance levels under meta-sparsity (although there is a significant standard deviation). This discrepancy is likely due to the nature of $T_1$ as the sole pixel-level classification task among others that are pixel-level regression tasks, suggesting that sparse meta-parameters might not be well-suited to support $T_1$ effectively. Overall, it can be summarized that the relatedness of the tasks influences their feature requirements, which in turn dictates both the sparsity pattern and performance outcomes. The competitive and cooperative dynamics of MTL might cause some tasks to struggle or perform poorly in the ensemble while others may excel. It emphasizes the significance of cautiously choosing and balancing tasks in an ensemble to enhance overall performance and sparsity efficiency.

Table 2: This table compares the performance of the tasks for the celebA dataset when a new unseen task is added during the meta-testing stage. The new tasks are highlighted in blue.

| Tasks | | Ssegmentation | male | smile | biglips | highcheek bones | wearing liptick | bushy eyebrows | Parameter sparsity |
|---|---|---|---|---|---|---|---|---|---|
| Meta-training | Meta-testing | $T_1$ | $T_2$ | $T_3$ | $T_4$ | $T_5$ | $T_6$ | $T_7$ | |
| | (finetuning +test) | IoU(↑) | Accuracy(↑) | | | | | | (%) |
| $T_1$-$T_7$ | $T_1$-$T_7$ | 0.9007±0.0012 | 0.9669±0.0014 | 0.9014±0.0043 | 0.6330±0.1680 | 0.8235±0.0104 | 0.9068±0.0165 | 0.6885±0.0936 | 43.6893±0.3593 |
| $T_1,T_2,T_3,T_7$ | $T_1,T_2,T_3,T_7$ | 0.9034±0.0013 | 0.7386±0.0188 | 0.5261±0.0010 | - | - | - | 0.6223±0.0051 | 30.0187±0.0011 |
| $T_1,T_2,T_3,T_7$ | $T_1,T_2,T_3,T_7,$ + $T_4,T_5,T_6$ | 0.9019±0.0005 | 0.9559±0.0191 | 0.8995±0.0116 | 0.6330±0.0130 | 0.8206±0.0066 | 0.9073±0.0070 | 0.7250±0.0890 | 30.0187±0.0011 |
| $T_1,T_2,T_3,T_7$ | finetune only $T_4,T_5,T_6$ | - | - | - | 0.6330±0.0011 | 0.7373±0.0117 | 0.8812±0.0096 | - | 30.0187±0.0011 |
| $T_2$-$T_7$ | $T_2$-$T_7$ | - | 0.9004±0.0203 | 0.7618±0.1602 | 0.6330±0.0072 | 0.7562±0.1109 | 0.8703±0.0154 | 0.6210±0.0001 | 30.2145±0.0016 |
| $T_2$-$T_7$ | $T_2$-$T_7$, +$T_1$ | 0.8985±0.0027 | 0.9615±0.0051 | 0.8977±0.0072 | 0.6330±0.0091 | 0.8278±0.0057 | 0.9102±0.0144 | 0.7529±0.0199 | 30.2145±0.0016 |
| $T_2$-$T_7$ | finetune only $T_1$ | 0.9033±0.0019 | - | - | - | - | - | - | 30.2145±0.0016 |

Table 3: This table compares the performance of the tasks for NYU dataset when a new unseen task is added during the meta-testing stage. The new tasks are highlighted in blue

| Tasks | | Segmentation $T_1$ | Depth est. $T_2$ | SN est. $T_3$ | Edge det. $T_4$ | Parameter sparsity |
|---|---|---|---|---|---|---|
| Meta-training | Meta-testing (finetuning+test) | IoU($\uparrow$) | MAE($\downarrow$) | CS($\uparrow$) | MAE($\downarrow$) | (%) |
| $T_1, T_2, T_3, T_4$ | $T_1, T_2, T_3, T_4$ | $0.2923 \pm 0.0101$ | $0.1397 \pm 0.0037$ | $0.7414 \pm 0.0055$ | $0.1395 \pm 0.0021$ | $44.1159 \pm 0.3696$ |
| $T_1,T_2,T_3$ | $T_1,T_2,T_3$ | $0.2937 \pm 0.0091$ | $0.1409 \pm 0.0035$ | $0.7460 \pm 0.0073$ | - | $43.4391 \pm 0.1925$ |
| $T_1,T_2,T_3$ | $T_1,T_2,T_3,+T_4$ | $0.3003 \pm 0.0067$ | $0.1389 \pm 0.0030$ | $0.7438 \pm 0.0005$ | $0.1404 \pm 0.0015$ | $43.4391 \pm 0.1925$ |
| $T_1,T_2,T_3$ | finetuning only $T_4$ | - | - | - | $0.1431 \pm 0.0020$ | $43.4391 \pm 0.1925$ |
| $T_2,T_3,T_4$ | $T_2,T_3,T_4$ | - | $0.1412 \pm 0.0040$ | $0.7402 \pm 0.0080$ | $0.1386 \pm 0.0019$ | $43.7577 \pm 0.1886$ |
| $T_2,T_3,T_4$ | $T_2,T_3,T_4,+T_1$ | $0.2950 \pm 0.0069$ | $0.1378 \pm 0.0020$ | $0.7444 \pm 0.0017$ | $0.1386 \pm 0.0003$ | $43.7577 \pm 0.1886$ |
| $T_2,T_3,T_4$ | finetuning only $T_1$ | $0.2828 \pm 0.0392$ | - | - | - | $43.7577 \pm 0.1886$ |

**Evaluating the efficacy of sparse backbone:** We study the effectiveness of the sparse backbone by introducing novel, previously unseen tasks during the meta-testing stage, i.e., a task different from the ones the model is meta-trained on. So, both the task and the data during meta-testing are unseen. We studied the performance of the sparse shared backbone network under two distinct scenarios: firstly, when the new task is integrated alongside the tasks from the meta-training phase, and secondly, when the model is fine-tuned exclusively on the new task. Note that the level of sparsity achieved during the meta-training stage is maintained during meta-testing by masking the layers that were zeroed out. Tables 2 and 3 show the performance of the new tasks for the celebA and NYU datasets, respectively. For comparison, we also show the performance of the meta-trained tasks during meta-testing.

For the celebA dataset, when tasks $T_4, T_5, T_6$ are added to the pre-existing set of tasks $T_1, T_2, T_3, T_7$ for meta-testing, the performance across the tasks improves or remains stable, without any significant degradation. Notably, task $T_3$ (smile classification) shows significant improvement with the inclusion of mouth-related tasks $T_4$ (big lips/no-big lips) and $T_6$ (wearing lipstick/not wearing lipstick) due to more focused learning on mouth features. Similarly, tasks $T_2$ (male classification) and $T_7$ (bushy eyebrows classification) benefit from this refined feature extraction, enhancing performance. Similar observations can be made when semantic segmentation $T_1$ (i.e., a pixel-level task) is added to the mix of classification tasks ($T_2 - T_7$). The segmentation task demands pixel-level, fine-grained image understanding, which enhances the shared feature representation, benefiting the classification tasks with more robust and discriminative features. For the NYU dataset (see Table 3), in both the cases, i.e., addition of $T_1$ and $T_4$, the performances of the tasks are consistent when compared to the outcomes of other settings. When these tasks are fine-tuned in a multi-task setting along with other tasks, there is either slight improvement or maintenance of performance levels for the existing tasks without any observed degradation.

These observations highlight the advantages of MTL, where simultaneous training on multiple tasks can lead to better overall performance through shared insights and learning dynamics. It also highlights the performance consistency across tasks, i.e., the sparse backbone demonstrates robust behavior even when new tasks are introduced in the mix. Occasionally, it is noticed that the addition of new tasks might also enhance the performance of meta-training (old) tasks. This suggests that the tasks mutually support and improve learning. Furthermore, the fact that meta-training tasks maintain stable performance during meta-testing, even with the introduction of new tasks, suggests that the proposed method is robust and may help prevent negative information transfer.

**Comparisons with other sparsity baselines:** To give a baseline performance comparison of the proposed meta-sparsity approach with other sparsification methods, Table 4 presents a comparison in three levels: (i) Sparsification approaches illustrated in Figure 1, (ii) Sparsity patterns are also referred to as masks in this discussion, and (iii) Parameter initialization before sparsification. For a fair comparison between the approaches, we forced all the sparsification methods to maintain the same % parameter sparsity achieved by meta-sparsity. From Table 4 it is evident that there is no one approach which works the best for all the tasks. The lowest magnitude parameter elimination for progressive and iterative sparsification shows promising results for $T_1, T_2, T_3$, but for $T_4$ the performance is suboptimal. Zeroing out the parameters randomly also shows some promising performance. However, it is to be noted that these performances are at

Table 4: Performance analysis of the sparsification approaches presented in Figure 1, for the NYU-v2 datset.

| Sparsification approaches | Sparsity patterns | parameter init. before sparsification | Segmentation T1 IoU(↑) | Depth est. T2 MAE(↓) | SN est. T3 CS(↑) | Edge det. T4 MAE(↓) |
|---|---|---|---|---|---|---|
| One-shot | Mask-1, magnitude | MTL dense | $0.2345 \pm 0.0008$ | $0.1777 \pm 0.0097$ | $0.6601 \pm 0.0250$ | $0.3069 \pm 0.0325$ |
| | Mask-3, random | MTL dense | $0.3185 \pm 0.0010$ | $0.1337 \pm 0.0018$ | $0.7412 \pm 0.0027$ | $0.1318 \pm 0.0026$ |
| | Meta mask | MTL dense | $0.2308 \pm 0.0106$ | $0.1682 \pm 0.0065$ | $0.6897 \pm 0.0062$ | $0.3036 \pm 0.0214$ |
| Iterative | Mask-2, iterative magnitude | MTL dense | $0.3082 \pm 0.0094$ | $0.1367 \pm 0.0014$ | $0.7346 \pm 0.0043$ | $0.1353 \pm 0.0008$ |
| | Meta mask | MTL dense | $0.3008 \pm 0.0080$ | $0.1373 \pm 0.0007$ | $0.7314 \pm 0.0027$ | $0.1356 \pm 0.0008$ |
| Progressive | Mask-2, iterative magnitude | random | $0.3855 \pm 0.0139$ | $0.1491 \pm 0.0209$ | $0.7800 \pm 0.0102$ | $0.1957 \pm 0.0161$ |
| | Mask-3, random | random | $0.3740 \pm 0.0155$ | $0.1361 \pm 0.0040$ | $0.7772 \pm 0.0079$ | $0.2176 \pm 0.0228$ |
| | Meta mask | random | $0.4018 \pm 0.0082$ | $0.1391 \pm 0.0007$ | $0.7877 \pm 0.0060$ | $0.1978 \pm 0.0017$ |
| Sparse training | Mask-1 , magnitude | random | $0.2918 \pm 0.0112$ | $0.1488 \pm 0.0049$ | $0.7202 \pm 0.0058$ | $0.2289 \pm 0.0035$ |
| | Mask-3, random | random | $0.2959 \pm 0.0077$ | $0.1530 \pm 0.0009$ | $0.7180 \pm 0.0033$ | $0.2389 \pm 0.0105$ |
| | Meta mask | random | $0.3906 \pm 0.0087$ | $0.1282 \pm 0.0050$ | $0.7827 \pm 0.0022$ | $0.2001 \pm 0.0102$ |
| Meta-sparsity | Meta-mask/pattern | random | $0.2923 \pm 0.0101$ | $0.1397 \pm 0.0037$ | $0.7414 \pm 0.0055$ | $0.1395 \pm 0.0021$ |

*Note:* Mask-1, one-shot sparsification by eliminating the lowest magnitude weights(Janowsky (1989)).
Mask-2, iterative magnitude sparsification by eliminating the lowest magnitude weights in steps.
Mask-3, random sparsification.
Meta-mask is the optimal sparsity pattern/mask learned by the meta-sparsity experiment.
The % parameter sparsity is kept constant across all the experiments, equal to the meta-sparsity achieved, i.e.,∼44.11%.

an optimal sparsity budget(∼44%) learned by meta-sparsity. As the above table shows, meta-sparsity achieves comparable performance across all tasks. The strength of our approach lies in its ability to dynamically (meta) learn sparsity patterns, which leads to an optimal amount of sparsity. Other methods require a sparsity budget, thresholds, or a sparsity step in case of iterative and progressive sparsification to regulate the level of sparsity, and very often, it is very tedious to find the correct balance between sparsity percentage and task performance. To verify the viability of the learned meta-sparsity patterns, we applied the learned meta-mask across various sparsification approaches. The meta mask/pattern consistently performed well for almost all tasks and across all approaches, demonstrating that the learned pattern is an optimal sparsity pattern for various tasks in an MTL setting.

**Other sparsity metrics:** In the context of sparse models, compression ratio[5] (CR) and speed-up[6] (Sp) are two important metrics that quantify the application of sparsity(Blalock et al. (2020)). The compression ratio measures the extent of model compression by comparing the size (in a number of parameters) of the original model with the compressed model. Speed-up is not a direct measure of the size reduction; it is a result of model compression. Due to sparsity, when a model is compressed, it typically requires fewer computational resources i.e., FLOPs, which can lead to faster processing times. Therefore, speed-up is an indirect measure of the operational efficiency of a model achieved due to model compression.

Figure 5 shows the task-wise compression ratio and speed-up achieved due to the fixed sparsity and meta sparsity for different task combinations. The meta-sparsity metrics are circled in red. Except for task $T_1$ (segmentation), these figures illustrate a trade-off between performance and both compression and speed. This observation aligns with our earlier discussion on parameter sparsity vs performance in the comparative performance analysis subsection. For example, in task $T_2$ (depth estimation), models with meta-sparsity achieve a computation speed of approximately $1.6\times$ faster than those of the dense backbone network. While fixed-sparsity models may be faster and more compressed than their meta-sparsity counterparts, this advantage often results in a trade-off with performance. For the segmentation task, it is apparent that meta-sparsity fails to boost performance and does not meet the anticipated outcomes. While the fixed sparsity models achieve notably greater compression, they can increase their speed by fourfold and exceed expected performance levels.

**Sparsity profiles:** Sparsity profiles illustrate the dynamic changes in the proportion of zero or inactive parameters within a model during the training process, effectively mapping the evolution of model compactness. Figure 6 illustrates the parameter sparsity profiles for the fixed and meta-sparsity experiments during

---

[5]$CR = \dfrac{total\ parameters}{no.\ of\ nonzero\ parameters}$

[6]$Sp = \dfrac{total\ FLOPs}{no.\ of\ nonzero\ FLOPs}$; FLOPs = floating point operations

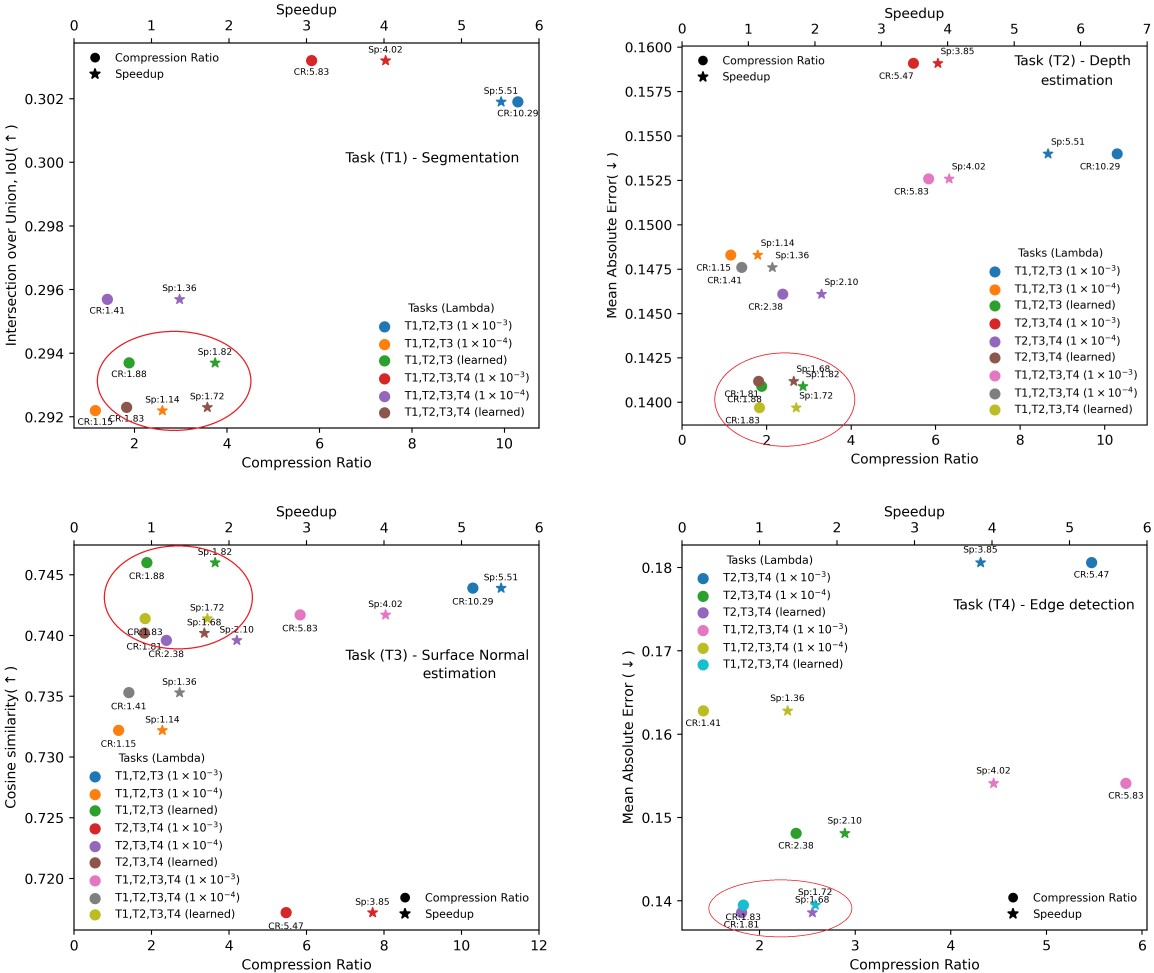

Figure 5: This figure presents a comparison of performance against both the compression ratio and speed-up across all tasks within the NYU dataset, with meta-sparsity experiments distinctly marked by red circles. It is important to note that the plots feature dual x-axes, with the lower axis denoting the compression ratio and the upper axis indicating the speed-up

training. It is evident from Figure 6(a) and 6(b) that sparsity increases sharply early in training and then plateaus, suggesting that sparsity is introduced quickly and maintained throughout the training process. The experiments with $\lambda = 1 \times 10^{-4}$ attain lower levels of parameter sparsity as compared to the $\lambda = 1 \times 10^{-3}$; this is obvious since $\lambda$ defines the strength of sparsity. Moreover, sparsity is achieved more swiftly in single-task setups than in multi-task ones. This is because MTL needs more iterations to determine which features are unnecessary for the ensemble and may be removed.

In Figure 6(c), which shows the sparsity profile for meta-sparsity settings, the sparsity levels appear more stable and less variable over iterations than fixed sparsity settings. The lines are relatively flat, indicating that once the sparsity level is set, it doesn't change much during the training for a very long time. This could suggest that meta-sparsity leads to a more consistent sparsity pattern, resulting in the consistent performance of the sparse model. A stable sparsity pattern may indicate that the model has learned a general representation that is not overly fitted to the noise or eccentricity of the training data, which can lead to better generalization on unseen data (or tasks).

**Structured vs unstructured sparsity:** In this work, we have focused on meta-sparsity within the context of a group (channel-wise) or structured sparsity. We propose that meta-sparsity is a versatile concept,

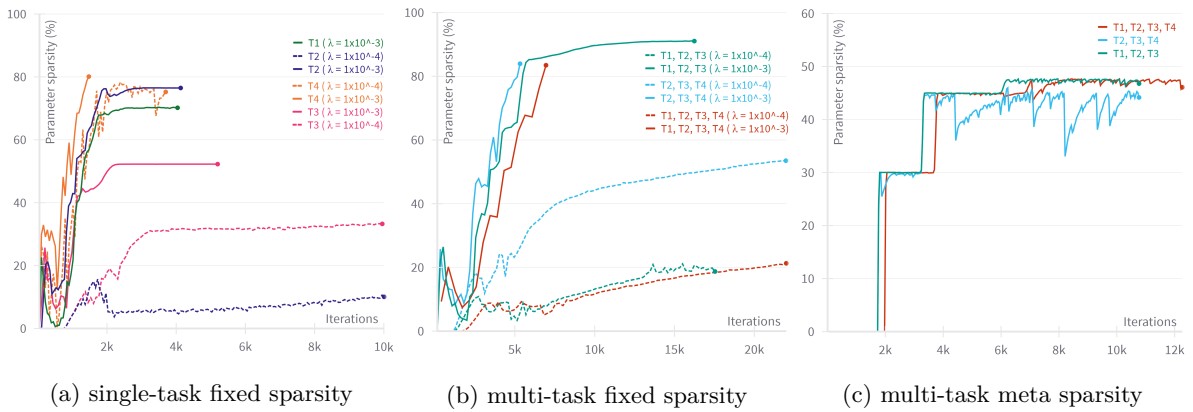

(a) single-task fixed sparsity  (b) multi-task fixed sparsity  (c) multi-task meta sparsity

Figure 6: Parameter sparsity patterns while training for (a) single-task fixed sparsity, (b) multi-task fixed sparsity, and (c) multi-task meta sparsity experiments. In the above plots, T1 represents semantic segmentation, T2 represents depth estimation, T3 represents surface normal estimation, and T4 represents edge detection tasks. In the fixed sparsity plots i.e., (a) and (b), the dotted lines represent the experiments where $\lambda = 1 \times 10^{-4}$, while the solid lines represent the sparsity pattern when $\lambda = 1 \times 10^{-3}$.

extendable to various forms of sparsity—be it structured, unstructured, penalty-driven, or any other pruning methods controlled by hyperparameters (denoted by $\lambda$ in this case). To support our claim, we extended the use of meta-sparsity to include fine-grained $l_1$ (unstructured) sparsity under various experimental scenarios: single-task fixed sparsity, multi-task fixed sparsity, and multi-task meta-sparsity.

Figure 7 shows the sparsity profile under $l_1 - l_2$ and $l_1$ type penalty-based sparsity settings. Figures 7(a) and (b) specifically address fixed sparsity scenarios, where the comparison reveals that models under $l_1$ sparsity achieve sparsity levels comparable to those subjected to group sparsity $(l_1 - l_2)$. The increase of parameter sparsity in models with $l_1$ regularization follows a linear steady pattern. In contrast, models with $l_1 - l_2$ regularization exhibit initial volatility in parameter sparsity, characterized by quick increases or variations. This is mainly because parameter groups are collectively deactivated in the structured sparsity approach. The performance of the tasks under structured and unstructured sparsity is mostly similar, with the exceptions of a few tasks performing well in a structured setting; see Tables 1 and 2 in the Appendix for a detailed comparison of the task-wise performance. In the case of the meta-sparsity setting i.e., Figure 7(c), it is

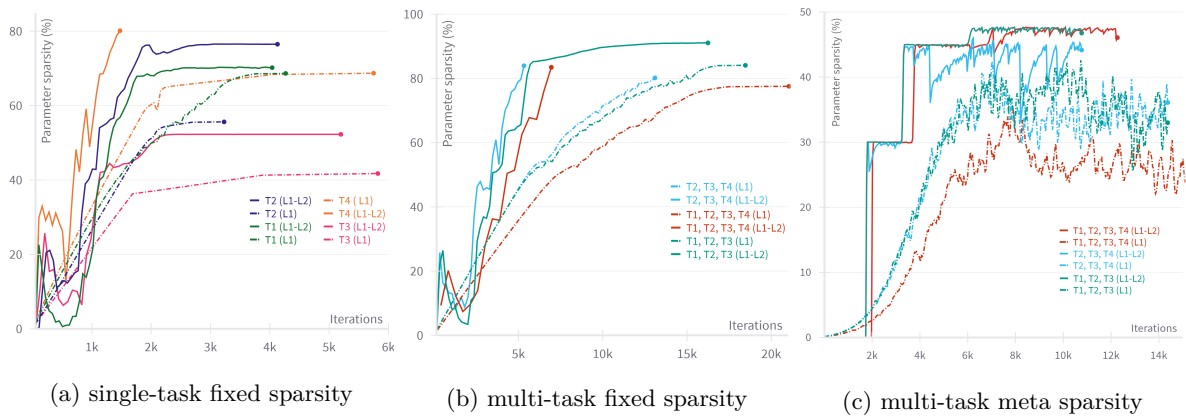

(a) single-task fixed sparsity  (b) multi-task fixed sparsity  (c) multi-task meta sparsity

Figure 7: Parameter sparsity patterns for structured ($l_1 - l_2$ channel-wise group sparsity, solid lines) and unstructured sparsity (fine-grained $l_1$ sparsity, '_.' lines) while training for (a) single-task fixed sparsity, (b) multi-task fixed sparsity, and (c) multi-task meta sparsity experiments. In the above plots, T1 represents semantic segmentation, T2 represents depth estimation, T3 represents surface normal estimation, and T4 represents edge detection tasks for the NYU dataset.

evident that the structured sparsity gives stable sparsity patterns, while those for unstructured sparsity

Table 5: Average performance for the four task combination i.e., $(T_1, T_2, T_3, T_4)$ of the NYU dataset under the meta-sparsity setting for various values of the regrowth parameter, $r_p$. The regrow parameter, say $r_p = x$, signifies that there is a $x\%$ chance that any sparsified channel will be regrown or reintroduced. In this work, we use Xavier initialization (Glorot & Bengio (2010)) to set the new values of the regrown filters. $r_p = 0$ represents the case when none of the parameters is re-grown during meta-training.

| Regrow prob. $r_p$ | Segmentation $T_1$ IoU($\uparrow$) | Depth est. $T_2$ MAE($\downarrow$) | SN est. $T_3$ CS($\uparrow$) | Edge det. $T_4$ MAE($\downarrow$) | Parameter sparsity (%) |
|---|---|---|---|---|---|
| 0 (no regrow) | 0.2923 | 0.1397 | 0.7414 | 0.1395 | 44.1159 |
| 0.2 | 0.3177 | 0.1339 | 0.7514 | 0.1398 | 30.1157 |
| 0.4 | 0.3100 | 0.1409 | 0.7437 | 0.1405 | 30.0186 |
| 0.6 | 0.3137 | 0.1356 | 0.7500 | 0.1382 | 29.9991 |
| 0.8 | 0.3170 | 0.1351 | 0.7472 | 0.1409 | 31.0982 |

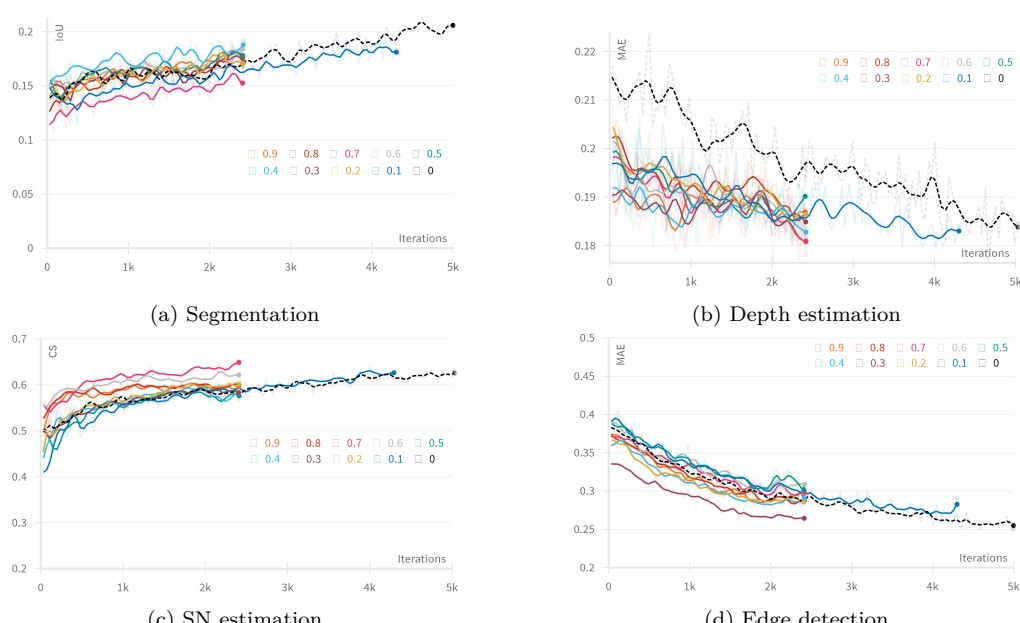

(a) Segmentation  (b) Depth estimation

(c) SN estimation  (d) Edge detection

Figure 8: Comparison of the no-regrow (for $r_p = 0$, in black dotted lines) vs regrow (for $r_p = 0.1, 0.2, 0.3,.., 0.9$) on the performance of the tasks during meta-training.

are highly variable with too many fluctuations. The stability could be beneficial in practical applications where consistent performance is crucial. The variability in unstructured sparsity may reflect a continuous adaptation to the learning task, which could be advantageous in non-stationary environments where the model needs to constantly adjust.

Overall, we show that the proposed approach of meta-sparsity can learn both structured and unstructured sparsity. However, the choice between these forms of sparsity depends on the specific requirements of the use case at hand. While unstructured sparsity often leads to greater model compression by zeroing out more parameters, this may come at the cost of network performance (Hoefler et al. (2021)). Structured sparsity, on the other hand, is better suited to the needs of current hardware designs. By zeroing out entire channels or filters, the sparse network architecture can be implemented more effectively on the current hardware accelerators like GPUs or specialized ASICs. When combined with hardware acceleration, structured sparsity can significantly reduce computation time. On the other hand, unstructured sparsity may not result in computational speedups without specific hardware support since the remaining non-zero values are dispersed throughout the matrix, prohibiting the efficient use of vectorized operations.

**Regrowing the parameters:** As discussed earlier, regrowing refers to the process of systematically reintroducing the sparsified parameters, which allows the network to recover from over-sparsification. While

this concept has been outlined in Section 3, it should be noted that the empirical results presented did not involve any regrowth (i.e., $r_p = 0$). The rationale behind this was to focus our analysis on evaluating the effectiveness of the proposed meta-sparsity approach. Adding regrowth would require adjusting another hyperparameter, which could affect the clarity of our findings on meta-sparsity.

Figure 8 illustrates the subtle impact of the regrowth rate on the meta-training phase of the different tasks. It is evident from the plots that the presence of regrowth leads to faster convergence, which indicates that reintroducing the parameters helps the model quickly find the optimal parameters. Although there are not very significant performance gains during meta-training. The test performance of some of these models that incorporated regrowth is shown in Table 5, along with the % parameter sparsity. This table compares the no-regrow setting to the regrow settings with $r_p = 0.2, 0.4, 0.6, 0.8$. Analysis of the table reveals minimal performance improvements observed with different regrowth rates, except for the segmentation task. However, the model with regrowth is relatively less sparse than the one without regrowth. Figure 8 and Table 5 suggest that an optimal regrowth probability may exist, which balances the trade-offs between model compression and learning efficacy.

# 6 Conclusion and future scope

This study demonstrates that model sparsity can be a learnable attribute rather than a feature determined by heuristic hyperparameters. Our proposed framework for learned sparsity, also referred to as *meta-sparsity*, shows that models may be (meta-)trained to naturally adopt sparse structures, eliminating the need for manual tuning of sparsity levels. Our outcomes show that this strategy can produce models that are not only efficient and compact but also perform well on a variety of tasks. Although the segmentation task for the NYU dataset, in particular, poses a challenge due to its inherent complexity, such as dense pixel maps of around 40 segmentation classes, the meta-sparsity framework has proved robust. This study especially focused on the MAML framework within the context of meta-learning. However, it is acknowledged that various advanced extensions of MAML or similar gradient-based meta-learning algorithms can potentially improve performance and are viable alternatives for implementation in this work. The aim of this study extends beyond merely enhancing the performance of the tasks; it proposes a concept that could lead to the development of parsimonious models.

Theoretically, the proposed meta-sparsity approach is versatile and can be applied to any number and types of tasks, as well as various types of sparsity. Based on the experiments, we have identified a few scenarios where the method may prove to be highly effective-

- *Similar or closely related tasks:* The definition of similar tasks is very subjective; however, in the context of this work, they can consider them as tasks that may require some similar set of features. Our experiments indicate that meta-sparsity performs best when the tasks are similar or closely related, like depth estimation, surface normal estimation, segmentation, and edge detection for the NYU-v2 dataset. In such cases, the approach is better able to identify and leverage optimal shared sparse patterns that benefit all tasks involved. When tasks are highly diverse, it may be more challenging to find an optimal shared sparse pattern that effectively supports all the tasks. In such cases, the variability in the task feature requirement can make it difficult to achieve good performance for all the tasks. The performance of the task combination $T_1, T_2, T_3, T_7$ for celebA is one such example.

- *Inclusion of pixel-level tasks:* Adding a pixel-level task to the task mix can enhance the performance of all tasks, as observed in our experiments with the CelebA dataset ($T_1 - T_7$ vs $T_2 - T_7$, here $T_1$ is the only pixel level task). Pixel-level tasks require more granular features, which may also benefit image-level tasks by providing richer feature representations.

Our approach represents a step towards developing *black box* sparsity, i.e., allowing models to learn an optimal sparsity pattern. This approach is not limited to a specific model, task, or type of sparsity. It is strengthened by the concept of meta-learning, which learns the sparsity pattern across a range of tasks and allows easy integration of new tasks to the sparse models. MTL facilitates the joint training of diverse tasks within a

single model. When combined with sparsity, MTL not only helps with model compression but also enhances the task's performance by effective feature sharing between the tasks.

In continuation of this work, various directions could be pursued. A few of them are listed below:

*Structured sparsity in diverse architectures:* Although structural sparsity has demonstrated potential in networks with residual connections like ResNet architectures, its applicability to other networks remains unclear. Future research might examine how different types of structured or block sparsity could be customized for different network architectures.

*Task transference in sparse models:* Exploring task transference in sparse multi-task models is a promising direction for further research. Investigating the sparse features that promote better sharing across tasks could provide new insights into multi-task networks and the impact of sparsity on cooperative learning.

*Hardware efficiency of sparse models:* It is also necessary to investigate how sparse models align with present and upcoming hardware capabilities. The aim is to translate model compression due to sparsity into tangible computational efficiency and speed improvements on different hardware platforms.

### Acknowledgments

Acknowledgments will be stated after the double-blind review process.

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
