# OpenReview forum: "Meta-Sparsity: Learning Optimal Sparse Structures in Multi-task Networks through Meta-learning"
_TMLR — Rejected by TMLR_

### Review · Reviewer_bH4d · 2024-06-26

**Summary Of Contributions:**

The paper introduces a novel method to address the problem of model compression. Specifically, it incorporates model sparsity into the training process, treating the degree of sparsity as a learning parameter. The primary idea involves adding a penalty-based term to the optimization process and dynamically learning the model and sparsity using a bilevel framework. To demonstrate the method's superiority, experiments are conducted on multi-task learning scenarios, where meta-learning episodes are typically similar or homogeneous.

**Audience:**

Yes

**Broader Impact Concerns:**

No impact concerns.

**Claims And Evidence:**

Yes

**Requested Changes:**

Requested Changes:

1.Add additional experiments as above mentioned.
2.Provide more discussion on the weakness 3 and 4.

**Strengths And Weaknesses:**

trengths:

1.The paper is well-written and well-organized. The graphs and footnotes enhance the reviewer's understanding, particularly the organization on the third page.

2. The proposed method is promising. By dynamically learning the sparsity parameter instead of setting it as a hyperparameter, the meta-sparsity approach may perform better in multi-task settings.

3. The experiments conducted are good, validating their method in some way.

Weaknesses:

1. The experiments lack a baseline comparison. How does the proposed meta-sparsity method compare with the methods presented in Figure 1?

2. The goal of sparsity is to reduce computational cost and memory usage. The experiment should provide more results and evaluations on these aspects.

3. How to understand D_query and D_sup, why must their intersection be zero (Equation 3)?

4. Equation (8) differs from the cited paper [1]. Is this a typo or a modification? In their paper, [z] indicates max(0, z).

[1] Deleu, Tristan, and Yoshua Bengio. "Structured sparsity inducing adaptive optimizers for deep learning." arXiv preprint arXiv:2102.03869 (2021).

---

> ### Author Response · Authors · 2024-07-16
> **Reply to Reviewer bH4d [1/2]**
>
> Thank you very much for your thoughtful and detailed review of our paper. We greatly appreciate your recognition of our contributions and acknowledgment of the strengths of our work. In response to the weaknesses you mentioned, we provide a detailed explanation of each point below.
>
> Note that necessary changes in the manuscript will be made, and a new manuscript will be uploaded once feedback from all the reviewers is received (as per TMLR guidelines).
>
> >1. Baseline comparison
>
> Initially, this work intended to propose a research direction in the study of sparse networks, emphasizing the exploration of learning sparse patterns. However, we understand the importance of comparing our proposed meta-sparsity method with existing methods to provide a comprehensive evaluation. To address this, we have done additional experiments to include baseline comparisons with the methods presented in Figure 1. We will also incorporate these results and a detailed discussion in the revised manuscript.
> Table: Comparative performance analysis of all the sparsification approaches presented in Figure 1.
> | Sparsification   | Sparsity                      | parameter       | Segmentation  |   Depth est.  |    SN est.    |   Edge det.   |
> |------------------|-------------------------------|-----------------|:-------------:|:-------------:|:-------------:|:-------------:|
> |  approaches      | patterns                      |  init. before   |       T1      |       T2      |       T3      |       T4      |
> |                  |                               | sparsification  |     IoU(↑)    |     MAE(↓)    |     CS(↑)     |     MAE(↓)    |
> |                  |                               |                 |               |               |               |               |
> | One-shot         | Mask-1, magnitude             | MTL dense       | 0.2345±0.0008 | 0.1777±0.0097 | 0.6601±0.0250 | 0.3069±0.0325 |
> |                  | Mask-3,   random              | MTL dense       | 0.3185±0.0010 | 0.1337±0.0018 | 0.7412±0.0027 | 0.1318±0.0026 |
> |                  | meta mask                     | MTL dense       | 0.2308±0.0106 | 0.1682±0.0065 | 0.6897±0.0062 | 0.3036±0.0214 |
> | Iterative        | Mask-2,   iterative magnitude | MTL dense       | 0.3082±0.0094 | 0.1367±0.0014 | 0.7346±0.0043 | 0.1353±0.0008 |
> |                  | meta mask                     | MTL dense       | 0.3008±0.0080 | 0.1373±0.0007 | 0.7314±0.0027 | 0.1356±0.0008 |
> | Progressive      | Mask-2,   iterative magnitude | random          | 0.3855±0.0139 | 0.1491±0.0209 | 0.7800±0.0102 | 0.1957±0.0161 |
> |                  | Mask-3,   random              | random          | 0.3740±0.0155 | 0.1361±0.0040 | 0.7772±0.0079 | 0.2176±0.0228 |
> |                  | meta mask                     | random          | 0.4018±0.0082 | 0.1391±0.0007 | 0.7877±0.0060 | 0.1978±0.0017 |
> | Sparse training  | Mask-1 , magnitude            | random          | 0.2918±0.0112 | 0.1488±0.0049 | 0.7202±0.0058 | 0.2289±0.0035 |
> |                  | Mask-3,   random              | random          | 0.2959±0.0077 | 0.1530±0.0009 | 0.7180±0.0033 | 0.2389±0.0105 |
> |                  | meta mask                     | random          | 0.3906±0.0087 | 0.1282±0.0050 | 0.7827±0.0022 | 0.2001±0.0102 |
> | Meta-sparsity    | Meta-mask/pattern             | random          | 0.2923±0.0101 | 0.1397±0.0037 | 0.7414±0.0055 | 0.1395±0.0021 |
>
> - Mask -1, one-shot sparsification by eliminating the lowest magnitude weights   [4].
> - Mask -2, iterative magnitude sparsification by eliminating the lowest magnitude weights in steps.
> - Mask-3, random sparsification.
> - Meta-mask is the sparsity pattern/mask learned during meta-sparsification.
> - NOTE: % parameter sparsity is kept constant across all the experiments, equal to the meta-sparsity achieved, i.e., 44.11 %.
>
> We present the comparison in three levels: (i). sparsification approaches in Table 1, (ii) sparsity patterns, and (iii) parameter initializations before sparsity. As the above table shows, meta-sparsity achieves comparable performance across all tasks. The strength of our approach lies in its ability to dynamically (meta) learn sparsity patterns, which leads to an optimal amount of sparsity. Other methods require a sparsity budget, thresholds, or a sparsity step in case of iterative and progressive sparsification to regulate the level of sparsity, and very often, it is very tedious to find the correct balance between sparsity percentage and task performance. To verify the viability of the learned meta-sparsity patterns, we applied it across various sparsification approaches. The meta mask/pattern consistently performed well for almost all tasks and approaches, demonstrating that the learned pattern is an optimal sparsity pattern for various tasks in an MTL setting. Tasks like T1 and T3 seem to perform very well for some approaches, while T4 performance worsens. Overall, meta-sparsification gives balanced performance throughout all the tasks.

---

> ### Author Response · Authors · 2024-07-16
> **Reply to Reviewer bH4d [2/2]**
>
> >2.  computational cost and memory usage.
>
> Evaluating computational cost and memory usage are indeed important aspects of model sparsity. We would like to bring to your notice that our paper utilizes compression ratio (CR) and speed-up (Sp) metrics to evaluate model compression. According to the reference [2], CR and Sp are established metrics that effectively quantify a model's compression ability.  While we understand the reviewer's request for more results on computational cost and memory usage, we believe that the CR and Sp metrics inherently address these aspects for various experiments involving fixed and meta sparsity for different task combinations.  These are discussed in detail under the subheading ‘other sparsity metrics’ on page 18 and Figure 5.
>
> >3.  Explanation of D_query and D_sup.
>
> Meta-learning aims to train models to adapt to new tasks quickly. During the meta-training phase, multiple tasks are sampled from a task distribution. This means that the D_sup (support set) and D_query (query set) for a task are sampled. So, the support set consists of labeled samples that are used to train or fine-tune the model, it may be viewed as a training set for the inner loop of meta training that focuses on task-specific training. The query set consists of data samples that are used to evaluate the model’s performance after it has been trained on the support set.  This involves computing the loss or error on the query set, which gives a measure of the model’s performance on new, unseen examples for that task. The accumulated (query set) losses of multiple tasks are used to meta-update the model’s parameters in the outer loop.
>
> Important – Although the query set may seem analogous to the traditional test set, it serves a different purpose. In contrast to a traditional test set, which is only used during the final evaluation of the trained model, the loss of the query set is used to update the meta-parameters during the meta-training phase. The query set is an important part of the meta-training phase as it contributes to the model's ability to generalize across tasks by providing feedback for bi-level meta-optimization.
>
> The intersection of D_sup  and D_query should be zero (i.e., disjoint) to ensure no overlap between the training data used for adaptation and the evaluation data used to assess the adaptation.  Measuring the model's performance on unseen query examples provides a reliable measure of its ability to adapt to new data.  Also, by ensuring that there is no overlap between these sets, it prevents the model from memorizing specific examples, allowing for a more accurate evaluation of its ability to generalize to new data.
> > 4. Discussion on Equation (8)
>
> Equation 8 in our manuscript is just another (elaborate) way of presenting the closed form (given by [3]) of the proximal gradient descent.  Indeed, there is a minor typo in Equation 8; we missed the mention of $\alpha$ in the condition. The corrected equation is:
>
> $prox_{\alpha\mathcal{R}}(\theta^g) = \begin{cases}\left[1- \dfrac{\alpha \lambda \sqrt{n^g}}{||\theta^g||_2}\right]\theta^g & ;||\theta^g||_2 > \alpha\lambda \sqrt{n^g}\\\\
> 0 &  ;||\theta^g||_2 \leq \alpha\lambda \sqrt{n^g}\end{cases}$
>
> We thank you for pointing this out. More clarification on the equation from [1]:
> In [1], the notation $z_+ = max(0,z)$ defines the positive part or thresholding operation.  This means that for any real number z:
>
> -  If z is greater than or equal to 0, $z_+ = z$
>
> - If z is less than 0,  $z_+ = 0$
>
> This operation effectively "clips" the z value at zero, setting any negative values to zero while keeping positive values unchanged. In the context of the proximal operator i.e.,
> $prox_{\alpha h}(x) = [1-\dfrac{\alpha \lambda}{||x||_2} ] _{+} x$,
>  it considers that the multiplier x is non-negative.
> If $1- \dfrac{\alpha \lambda}{||x||_2}$ is negative, (i.e., if $||x||_2 \leq \alpha \lambda$) it gets equal to 0, so consequently zeroing out the entire vector x.
>
> This is how we have represented the proximal operator in the above equation (Equation 8 in our manuscript). Therefore, the two equations i.e., Equation 6 in [1] and the one above (corrected) are essentially the same, with the new equation providing a more detailed and explicit form, particularly for the group case. We will correct Equation 8 in the revised manuscript.
>
> References -
> [1] Deleu, T., & Bengio, Y. (2021). Structured sparsity inducing adaptive optimizers for deep learning. arXiv preprint arXiv:2102.03869.
>
> [2] Blalock, D., Gonzalez Ortiz, J. J., Frankle, J., & Guttag, J. (2020). What is the state of neural network pruning?. Proceedings of machine learning and systems, 2,129-146.
>
> [3] Combettes, P. L., & Wajs, V. R. (2005). Signal recovery by proximal forward-backward splitting. Multiscale modeling & simulation, 4(4),1168-1200.
>
> [4] Janowsky, S. A. (1989). Pruning versus clipping in neural networks. Physical Review A, 39(12), 6600.

---

### Review · Reviewer_1jvJ · 2024-07-29

**Summary Of Contributions:**

1. **Introduction of Meta-Sparsity Framework:** The paper presents a novel framework called meta-sparsity that leverages meta-learning to dynamically learn the optimal sparse structures in multi-task networks. This approach allows the network to determine the sparsity patterns dynamically rather than relying on manually tuned hyperparameters.

2. **Penalty-Based Structured Sparsity:** Inspired by Model Agnostic Meta-Learning (MAML), the method applies penalty-based, channel-wise structured sparsity during the meta-training phase, optimizing the shared parameters in a multi-task learning setting.

3. **Comprehensive Evaluation:** The effectiveness of the meta-sparsity framework is rigorously evaluated on two datasets, NYU-v2 and CelebAMask-HQ, across various tasks ranging from pixel-level to image-level predictions.

4. **Versatility in Sparsity Types:** While focusing on channel-wise structured sparsity, the approach is validated for unstructured sparsity as well, demonstrating its broad applicability.

**Audience:**

Yes

**Claims And Evidence:**

Yes

**Requested Changes:**

1. **Performance Stability Discussion:** The authors should provide a more detailed discussion on the reasons for the observed performance instability of the proposed method, particularly in the context of the identified weaknesses.

2. **Theoretical Justification:** Where possible, include theoretical proofs or arguments to explain the performance instability and to justify the conditions under which the proposed method performs well.

3. **Application Scenarios:** Clearly delineate the scenarios and contexts in which the proposed method is most applicable, based on both empirical evidence and theoretical considerations.

**Strengths And Weaknesses:**

## Strengths

1. **Innovative Approach:** Integrating meta-learning with sparsity constraints is a novel idea that addresses current problems in the existing studies.

2. **Comprehensive Experiments:** The authors conduct extensive experiments, using a variety of datasets to validate their approach. The results are compelling and show clear improvements over existing methods.

4. **Good Presentation:** The paper is well-structured and clearly written, making it easier for readers to understand and follow the proposed methods and results.

## Weaknesses

1. **Group Sparsity Effectiveness:** According to Figure 4, group sparsity does not seem to have a positive impact on the CelebAMask-HQ dataset. The "no sparsity" model appears to perform better overall across all metrics.

2. **Meta-Learning Stability:** Figure 3 indicates that meta-learning performance on the NYU-v2 dataset is not very stable. The variance in performance is quite high for tasks other than edge detection, indicating inconsistent results.

---

> ### Author Response · Authors · 2024-08-07
> **Reply to Reviewer 1jvJ**
>
> Thank you for thoroughly evaluating the strengths and weaknesses of our work and providing valuable feedback for improvement. In the following points we response to your concerns, and we will make suitable additions in the manuscript once the feedback from all the reviewers is received.
>
> > On Performance stability
>
> - Regarding the stability of the task performances for the proposed meta-sparsity approach, particularly in the context of the NYU-v2 dataset (Figure 3 in manuscript), Please refer to Table 1 in the supplementary material (which forms the basis of Figure 3), which quantitatively demonstrates that the standard deviations are consistently low. To enhance the visibility and comparability of the performance differences, we chose a narrow y-axis range for the plots in Figure 3, which may have amplified the visual impact of the variability of the error bars representing the standard deviation.
> - However, it is evident that for many tasks the variance in the performance of meta-sparsity is slightly greater than the rest (no sparsity and fixed sparsity). One probable reason can be ‘Regularization Stochasticity’.  It is the variability in the learned sparsity patterns because of the stochastic nature of (mini-batch) gradient-based optimization and the dynamic penalty-based sparsity applied during training.  This stochasticity can result in different paths of convergence and patterns of sparsity when the same experiment is run multiple times; therefore, the variability in the performance.
> Another factor to consider, that can amplify these stochastic effects, is the random initialization of the strength of regularization hyperparameter ($\lambda$), which is trainable/ learned in meta-sparsity.  We sample the initial value of $\lambda$ from a uniform distribution between 0.1 and 1. Although we aim to limit the randomness, this range can still cause variations in the paths of convergence and the resulting patterns of sparsity.
>
> > Application Scenarios
>
> Theoretically, the proposed meta-sparsity approach is versatile and can be applied to any number and types of tasks, as well as various types of sparsity. Based on our experiments, we have identified scenarios where the method proves to be highly effective:
>
> - Similar and closely related tasks: The definition of similar tasks is very subjective; however, here we can consider them as tasks that may require some similar set of features. Our experiments indicate that meta-sparsity performs best when the tasks are similar or closely related, like depth estimation, surface normal estimation, segmentation, and edge detection for NYU dataset.  In such cases, the approach is better able to identify and leverage optimal shared sparse patterns that benefit all tasks involved.
> - Diverse task sets: when tasks are highly diverse, it may be more challenging to find an optimal shared sparse pattern that effectively supports all the tasks. In such cases, the variability in the task feature requirement can make it difficult to achieve good performance for all the tasks. The performance of the task combination T1, T2,T3,T7 for celebA  is one such example.  Also, the underperformance of task T7  in case of meta-sparsity experiments can be due to the same reason. This is also discussed in Section 5 under subheading “Assessing the viability of meta-sparsity" in the manuscript.
> - Inclusion of pixel level tasks : Adding a pixel-level task to the task mix can enhance the performance of all tasks, as observed in our experiments with the CelebA dataset (T1-T7 vs T2-T7, here T1 is the only pixel level task). Pixel-level tasks require more granular features, which can also benefit image-level tasks by providing richer feature representations.
>
> These insight can guide practitioners in selecting suitable contexts for applying the proposed method.

---

### Review · Reviewer_hTuS · 2024-08-17

**Summary Of Contributions:**

The paper addresses the challenge of learning shared and optimal sparsity in DNNs in a multi-task learning (MTL) setting. The proposed method, Meta-Sparsity, employs a bilevel optimization approach similar to Model Agnostic Meta-Learning (MAML), and meta-learns the optimal shared initialization parameters over multiple tasks under a channel-wise, structured sparsity penalty. The penalty strength is also meta-learned along with the model initialization during meta-training, thereby achieving meta-learnable dynamic sparsity. The method is evaluated on two MTL datasets, showing a good model-compression ratio (and inference speed-up) to performance trade-off on the NYU-v2 dataset.

**Audience:**

Yes

**Broader Impact Concerns:**

There are no obvious ethical implications of this work that need to be addressed.

**Claims And Evidence:**

No

**Requested Changes:**

### Critical:
- Please kindly refer to the weaknesses listed above.

### Minors
- Fig. 3 caption, regularization strength $10^{-}4,10^{-}3$ should be $10^{-4}, 10^{-3}$.
- Could the authors please give more details on the experimental settings and baselines? For example, do single-task/multi-task with fixed-sparsity also adopt meta-learning for MTL to meta-learn the model initialization ? If so, the subtle difference between the baselines and meta-sparsity lies in whether $\lambda$ is meta-learned?
- On page 13, Experimental setup. The authors stated: *for a fair comparison... all the experiments in this work use the same ... train-validation-test split, and hyperparameters*.
    - Do these experiment hyperparameters include those of the methods', e.g., multi-task + $\lambda= 1e^{-4}$ and multi-task + $\lambda=1e^{-3}$ would use the same value for learning rate?

**Strengths And Weaknesses:**

### Strengths

- The idea of meta-learning sparse structures with learnable sparse penalty strength through a bi-level optimization framework seems new.
- The proposed method demonstrates a favourable compute-to-performance trade-off on the NYU-v2 dataset during meta-testing.

$~$

### Weaknesses
- **Clarification on Section 3.4 Proposed approach - Meta-sparsity: I am confused about how $\lambda$ is meta-learned, particularly, what/how loss gradients are backpropagated to optimize $\lambda$ in the outer-loop.**
  - Eqn.(4) and Eqn.(11), along with the description on page 10 under "META-TRAINING" and Algorithm 1, all seem to suggest that the l1-l2 regularization term, hence $\lambda$, is *not involved* in the inner-loop objective for optimizing the task parameters $\Theta_{E_i}$ initialized from $\Theta_{meta}$.
  - If the inner-loop indeed does not involve $\lambda$, then there are no gradients w.r.t. $\lambda$ through the inner-loop during backward propagation of the outer-loop loss in Eqn.(10). Meanwhile, with the gradients of the outer-loop l1-l2 regularization term w.r.t. $\lambda$ *alone* (in Eqn.(10)), wouldn't $\lambda$ diverge towards negative infinity as the l1-l2 norm is always positive?
  - Given this, I am unsure on how meta-learning $\lambda$ is feasible within such a bi-level optimization setup. Could the authors kindly clarify these points and provide further insight into the mechanism by which $\lambda$ is meta-learned in the proposed framework?

$~$
- **More supporting evidence for the argument on meta-sparsity learns optimal sparsity without rely heavily on manual hyperparameter tuning.**

    - The authors seemingly made this claim in several places, for example.
      > - Abstract, "...unlike traditional sparsity methods that rely heavily on manual hyperparameter tuning"
      > - Conclusion, "This study demonstrates that model sparsity can be a learnable attribute rather than a feature determined by heuristic hyperparameters."
    - While making $\lambda$ in Eqn. (10) learnable indeed no longer requires manually setting it for meta-training, however, the learning dynamics of $\lambda$, hence the optimal sparsity, still depend on the meta-hyperparameters associated with $\lambda$, e.g., the outer-loop learning rate for $\lambda$, the initialization of $\lambda$.
    - Is the proposed method sensitive to meta-hyperparameter selections? Can the proposed meta-sparsity consistently achieve a better compute-to-performance trade-off when varying its meta-hyperparameters over reasonable ranges, compared to sparsity with manually selected $\lambda$ values.

$~$
-  **Experimental results of the proposed method on CelebAMask-HQ indicate some areas for improvement.**
    - Comparing the performance of meta-sparsity and the baselines is somewhat difficult when their sparsity levels are so different。
        -  While meta-sparsity outperforms $\lambda=1e^-4$ and $\lambda=1e^-5$ in some cases, the sparsity levels of these baselines ($<1$% for $\lambda=1e^-5$, and $>80$% for $\lambda=1e^-4$) are significantly different from that of meta-sparsity, which is around ~$30$% to $40$%. Will the proposed method still perform better compared to a baseline with approximately the same sparsity level (compression ratio) at the end of training on the CelebAMask-HQ dataset?
    - Has meta-sparsity really learned the optimal sparsity on CelebAMask-HQ?
        -  Since one major motivation for the proposed method is to learn optimal sparsity in MTL, could the authors potentially provide some insights on why the final learned sparsity of the proposed method is not close to 0% on the CelebAMask-HQ dataset, as multi-task learning with no sparsity (Blue) either outperforms or performs on par with the proposed method in all settings, so clearly no sparsity is the optimal sparsity in this MTL setting.
$~$
- **A lack of comparison to baselines in Tab. 2&3 to support the authors' claim on meta-sparsity's efficacy on unseen tasks.**
    - The authors claim that meta-sparsity is robust and effective on unseen tasks, as stated in several places (see quotes below) and use results from Tables 2 & 3 as supporting evidence (as these tables seem to be the only places where unseen task generalization results are shown):
      >- indicates the robustness of the proposed method and its capability to prevent negative information transfer. Pg. 13
      >- Robustness validation ... Pg.3
      >- The boarder objective of this work is to learn the sparsity patterns... that can be fine-tuned for the same or unseen tasks in the meta-testing stage.
      >
    - However, Tables 2 & 3 show results *only* for the proposed method.
    - Without a comparison to baselines, this analysis might seem somewhat less relevant and provides limited insights into the advantages of the proposed learnable sparsity versus non-learnable sparsity.
    - As a result, I am not sure whether I can deduce the same conclusion made by the authors that meta-sparsity is robust and effective on unseen tasks.

---

> ### Author Response · Authors · 2024-08-22
> **Reply to  Reviewer hTuS [1/3]**
>
> We appreciate the thoughtful and detailed comments from Reviewer hTuS. Your feedback was very helpful in refining our manuscript. Below is our response to the concerns you raised.
>
> > Clarification on Section 3.4 Proposed approach
>
> Thank you for your insightful observations. You are correct that the inner loop does not involve sparsification, and there are no gradients w.r.t.  $\lambda$ in the inner loop, as Equation 11 does not consist of a regularization term. The inner loop is focused only on episode-specific learning, where the goal is to adapt the (single-task/multi-task) parameters for each task or task-combination (i.e., episode).  This helps in capturing the nuances of individual episode without overfitting on the broader task distribution. While the outer loop adjusts the meta parameters (initial parameters) along with inducing sparsity based on its performance across a range of tasks (episodes). In a broader sense, the outer loop helps in extracting insights that are common and useful across different tasks. In the outer loop, the regularization term in Equation 10 consisting of  $\lambda$ causes sparsification along with appropriately updating the  $\lambda$ parameter value. This is an iterative process, the sparsity pattern in the backbone network determined by the outer loop, is preserved and carried forward into the next inner loop updates in the next epoch. Therefore, the outer loop’s role is to introduce and optimize parameter sparsity in the backbone network, while the inner loop trains various task combinations on the sparsified backbone.
>
> Preventing Divergence of  $\lambda$ - To avoid the potential divergence of  $\lambda$ towards negative infinity, we have applied the Softplus function to  $\lambda$, which ensures that it remains positive throughout the optimization process. The Softplus function, defined as $\text{Softplus}(x) = \frac{1}{\beta} \log(1 + \exp(\beta \cdot x))$, acts as a smooth approximation to the ReLU function and effectively constrains  $\lambda$ to positive values. By ensuring that  $\lambda$ remains positive, we avoid the issue of  $\lambda$ diverging towards negative infinity, hence ensuring that the regularization term continues to function as intended in balancing sparsity and model performance. We will clarify these points in the updated manuscript.
>
> > More supporting evidence for the argument on meta-sparsity learns optimal sparsity without relying heavily on manual hyperparameter tuning.
>
> For better understanding, below we clarify our claim, followed by an explanation related to the said weakness.
> Clarification of the claim - Our claim that the proposed meta-sparsity framework allows for learning model sparsity without relying on manual hyperparameter tuning specifically refers to  $\lambda$, the regularization parameter controlling the sparsity. By making  $\lambda$ a learnable parameter, we eliminate the need for manual tuning of this particular hyperparameter, allowing the model to adaptively learn the optimal level of sparsity during the training process.
>
> Meta-hyperparameters and their learnability – In this work, while we focused on making  $\lambda$ learnable, we acknowledge that the other meta-hyperparameters, such as the outer loop learning rate and initialization of $\lambda$, were not made learnable. This decision was intentional, as our primary goal is to investigate the effects of learning the sparsity causing parameter  $\lambda$ within the meta-sparsity framework. We recognize that other meta-hyperparameters could also be made learnable through meta-learning, and this presents an interesting direction of research.
>
> Sensitivity analysis - Regarding the sensitivity of our method to the meta-hyperparameters selections, we agree that varying these meta-hyperparameters could potentially influence the learning dynamics of  $\lambda$ and, consequently, the overall performance of the model. However, conducting a comprehensive sensitivity analysis of these meta-hyperparameters is beyond the scope of this work. All our experiments were designed with the same meta-hyperparameters to isolate and study the effects of learning $\lambda$ for both structured and unstructured sparsity. Consider, for example, a few key meta-hyperparameters such as the outer loop learning rate, initial value of $\lambda$, batch size, and number of inner loop steps. Conducting a sensitivity analysis across multiple values for these parameters, especially considering different task combinations and types of sparsity, would be a significant undertaking. This would require a detailed examination of how each combination impacts both sparsity and performance and extensive computational resources. We believe such an exhaustive analysis is not the primary focus of this work.

---

> > ### Author Response · Authors · 2024-08-22
> > **Reply to  Reviewer hTuS [2/3]**
> >
> > > Experimental results of the proposed method on CelebAMask-HQ indicate some areas for improvement. Comparing the performance of meta-sparsity and the baselines is somewhat difficult when their sparsity levels are so different。
> >
> > We understand that fixed sparsity and meta-sparsity result in different levels of sparsity, with meta-sparsity dynamically learning the optimal sparsity level during training. The core idea of our work is to eliminate the need for manually selecting a  $\lambda$ value that balances good performance with an appropriate level of sparsity. The CelebAMask-HQ dataset serves as an ideal example of this challenge: with  $\lambda = 10^{-5}$, the model achieved low sparsity, and with $\lambda = 10^{-4}$, it reached very high sparsity. However, in both cases, the performance was mostly suboptimal. Meta-sparsity, on the other hand, learns the optimal sparsity level and, for some task combinations, achieves performance close to that of a model with no imposed sparsity, however, with significantly fewer parameters.
> >
> > It is difficult to directly compare whether meta-sparsity would still perform better than baselines with approximately the same sparsity level, as the sparsity in meta-sparsity is learned dynamically. In our experiments, the optimal sparsity level found by meta-sparsity was typically around 30-40%, and we observed that once this level was reached, the amount of sparsity remained relatively stable or, say, did not exceed this (see Figure 6(c) for NYU dataset). Therefore, an evaluation of its performance at a sparsity level greater than the optimal sparsity achieved by meta-sparsity is not possible.
> >
> > > Has meta-sparsity really learned the optimal sparsity on CelebAMask-HQ?
> >
> > In this work, the meta-sparsity framework is expected to strike a balance between the model performance and sparsity, thereby reducing the unnecessary parameters while maintaining or sometimes improving the task performance. The fact that the learned sparsity is not 0% suggests that the method identified certain parameters as redundant across tasks and pruned them to induce sparsity. However, it is important to note that optimal sparsity is task combination (or task) dependent, as discussed in the manuscript. For example, the meta-sparsity performance of almost all the tasks in the task combination T1-T7 is at par with the no-sparsity performance, while for T1,T2,T3,T7 the performances are suboptimal. In the latter case, the non-performing tasks T2,T3,T7 which are image-level tasks, may require a broader range of features (less sparse or no sparse backbone), while T1, which is a pixel-level task, performs well with the meta-sparse backbone. The sparsity learned by our method reflects a compromise where some parameters are pruned, but a significant portion is retained in an attempt to preserve the performance.
> >
> > The observation that no sparsity experiments perform well in MTL setting suggests that, for the CelebAMask-HQ dataset, the tasks benefit from a denser network where more parameters are available for learning. This doesn't necessarily imply that sparsity is ineffective; rather, it indicates that in this specific setting, the trade-off between sparsity and performance leans more towards retaining more parameters. This outcome also highlights the importance of understanding the specific requirements of different datasets and task combinations when applying sparsity.
> >
> > > A lack of comparison to baselines in Tab. 2&3 to support the authors' claim on meta-sparsity's efficacy on unseen tasks.
> >
> > Clarification of our claims - The claims related to unseen tasks in the manuscript are intended to highlight that the meta-sparsity approach can generalize to tasks not included in the training phase. Specifically, we observed that when these unseen tasks are introduced during the testing phase, the model can perform at least similarly to how it would if those tasks had been included during (meta) training. We will soften the claims to make it clearer that these observations are made within the context of the meta-sparsity experiments alone.
> >
> > Regarding Baseline Comparisons- While we did not include baseline comparisons directly in Tables 2 and 3 of the main manuscript, it is important to note that the baseline results for single-task and multi-task settings are provided in Tables 1 and 3 of the supplementary material. It is evident that the unseen tasks perform on par with, or even better than, their single-task counterparts and multi-task performance in some cases. We focused on presenting the performance of the meta-sparsity approach only in Tables 2 and 3 to avoid confusion. Including a direct comparison between meta-sparsity and non-meta methods in these tables could have made it difficult to conduct a fair evaluation, as our primary goal was to compare the performance of tasks when they were meta-trained versus when they were not.

---

> > > ### Author Response · Authors · 2024-08-22
> > > **Reply to Reviewer hTuS [3/3]**
> > >
> > > > Requested changes - Minors
> > > Fig. 3 caption, ..
> > >
> > > Thank you for pointing this out, we will correct this in the caption of Figure 3.
> > >
> > > > Could the authors please give more details on the experimental settings and baselines? ...
> > >
> > > The single-task and multi-task experiments do not utilize meta-learning for model initialization; they follow standard single-task and multi-task settings. Meta-learning is only applied in the meta-sparsity experiments. We will update Section 4 of the manuscript to clarify this distinction. Additionally, we will include detailed experimental settings in the supplementary material and in our GitHub repository to ensure reproducibility.
> > >
> > > > On page 13, Experimental setup... Do these experiment hyperparameters include those of the methods, e.g., multi-task + and multi-task + would use the same value for learning rate?
> > >
> > > Yes, for all experiments—single-task, multi-task, with and without sparsity (both fixed and learned)—all hyperparameters, including the learning rate, were kept consistent.

---

### Author Response · Authors · 2024-08-22
**Revised manuscript uploaded**

Dear **Reviewers (hTuS, 1jvJ, bH4d)**,

We would like to sincerely thank all three reviewers for their insightful feedback and constructive suggestions. We truly appreciate the time and effort you invested in reviewing our work. Your feedback not only strengthened the current manuscript but also provided us with valuable directions for future research. Thank you for your thoughtful contributions.

Based on the reviews, we have uploaded the revised manuscript.

The following are the major updates-

1. Added Table 4, which includes a comparative performance analysis of the sparsification approaches, along with the related discussion.
2. Updated the text of Section 5, "Assessing the viability of meta-sparsity", explaining the performance stability and variations.
3. Discussed the application scenarios in the conclusion.
4. Minor corrections are done, and typos are fixed.

We also thank the **Action Editor** for considering our manuscript for the review process.

Thank you

---

### Decision · Action_Editor_KYm6 · 2024-11-06

**Recommendation:** Reject

**Comment:**

This paper proposes Meta-Sparsity, a novel framework that integrates meta-learning with structured sparsity for parameter optimization in multi-task learning.
- The method introduces dynamic sparsity, where sparsity patterns are learned rather than manually set, providing a promising approach for model compression.

However, concerns raised by reviewers limit its impact.
- The primary weaknesses include insufficient baseline comparisons, making it difficult to assess the true effectiveness of Meta-Sparsity. The method's performance stability, particularly on unseen tasks and across datasets, remains unclear, as variance in results suggests potential inconsistencies.
- Additionally, theoretical justification is limited, especially in explaining how meta-learning enhances task generalization.
- The reviewers also recommend further clarifications regarding performance stability, comprehensive baseline comparisons, and stronger theoretical support.

Overall, while the paper holds potential, these issues must be addressed before it can be recommended for acceptance.

**Audience:**

Yes. The paper introduces an innovative framework combining meta-learning and sparsity in multi-task learning, which could attract interest from TMLR’s audience focused on model compression and efficiency.

**Claims And Evidence:**

The submission introduces Meta-Sparsity, a framework that leverages bilevel optimization to dynamically learn optimal sparsity structures in multi-task learning (MTL) using a meta-learning approach inspired by MAML. Key strengths include the innovative integration of meta-learning with sparsity constraints and comprehensive experimentation across multiple datasets (e.g., CelebAMask-HQ, NYU-v2), showcasing improvements in parameter efficiency. However, significant weaknesses remain. The lack of fair comparisons to meta-learning baselines that meta-learn the initialization only, especially on unseen tasks, limits the ability to attribute observed gains solely to the proposed method. This makes it unclear whether the observed improvements in Meta-Sparsity over the baselines are due to the effectiveness of meta-learning the sparsity level or simply the benefits of meta-learning itself. Additionally, the framework lacks theoretical justification, and its claimed stability benefits are undermined by inconsistent performance results.

**Resubmission Of Major Revision:**

The authors may consider submitting a major revision at a later time.

---

> ### Author Response · Authors · 2024-11-12
> **About the final decision**
>
> We appreciate your insightful review and the detailed feedback concerning our manuscript.
>
> We acknowledge the decision and understand the concerns raised regarding our submission, particularly the need for more comprehensive baseline comparisons, more precise theoretical justifications, and improved performance stability assessments. Your comments have highlighted essential areas where we can further strengthen the manuscript.
>
> While we were hoping for reviewer comments post-rebuttal (on the updated manuscript) to incorporate updates directly into (this) manuscript, we will now take this opportunity to carefully respond to all feedback. Our goal is to significantly refine the manuscript, ensuring a robust theoretical foundation, more comprehensive experimentation with additional meta-learning baselines, and clearer explanations of performance trends.
>
> We remain committed to improving the quality of our work and look forward to resubmitting a stronger and more impactful version of the manuscript soon.
>
> Thank you again for your constructive feedback and for considering our work for TMLR.